# Mirror-image ligand discovery enabled by single-shot fast-flow synthesis of D-proteins

Alex J. Callahan[1,11], Satish Gandhesiri[1,11], Tara L. Travaline ®[2], Rahi M. Reja[1], Lia Lozano Salazar[1], Stephanie Hanna[1], Yen-Chun Lee[1,9], Kunhua Li[2], Olena S. Tokareva[2], Jean-Marie Swiecicki ®[2,10], Andrei Loas[1], Gregory L. Verdine ®[2,3,4,5], John H. McGee ®[2] ✉ & Bradley L. Pentelute ®[1,6,7,8] ✉

Widespread adoption of mirror-image biological systems presents difficulties in accessing the requisite D-protein substrates. In particular, mirror-image phage display has the potential for high-throughput generation of biologically stable macrocyclic D-peptide binders with potentially unique recognition modes but is hindered by the individualized optimization required for D-protein chemical synthesis. We demonstrate a general mirror-image phage display pipeline that utilizes automated flow peptide synthesis to prepare D-proteins in a single run. With this approach, we prepare and characterize 12 D-proteins – almost one third of all reported D-proteins to date. With access to mirror-image protein targets, we describe the successful discovery of six macrocyclic D-peptide binders: three to the oncoprotein MDM2, and three to the E3 ubiquitin ligase CHIP. Reliable production of mirror-image proteins can unlock the full potential of D-peptide drug discovery and streamline the study of mirror-image biology more broadly.

The biopolymers that carry out the central dogma of all known life are generated from a set of homochiral building blocks: D-deoxyribonucleic acids for DNA, D-ribonucleic acids for RNA, and L-amino acids for proteins. When strung together, these chiral monomers form three-dimensional structures with biological activities that are inherently chiral. Alternatively, unnatural biopolymers made from building blocks of the opposite chirality form mirror-image three-dimensional structures. These compounds retain the activity of their natural counterpart in the mirror-image biochemical space, but are not recognized nor degraded by natural biomolecules[1–4]. As a result, mirror-image biopolymers are useful tools to probe or perturb living systems. One such class of mirror-image compounds, D-peptides, are particularly valuable as therapeutic scaffolds, but remain under-developed.

Metabolic stability is a key bottleneck for the design of peptide-based therapeutics, and mirror-image peptides are uniquely resistant to degradation. The greater chiral and structural complexity of peptides make them valuable scaffolds to target protein surfaces not well-suited to disruption with small molecules[5]. Despite these benefits, unmodified canonical peptides are rapidly degraded in vivo, and chemical modification to extend their biological half-life is critical in the design of peptide-based drugs. Modern techniques for peptide drug discovery integrate stabilizing modifications early into design

[1]Department of Chemistry, Massachusetts Institute of Technology, 77 Massachusetts Avenue, Cambridge, MA 02139, USA. [2]FOG Pharmaceuticals Inc., 30 Acorn Park Drive, Cambridge, MA 02140, USA. [3]Department of Stem Cell and Regenerative Biology, Harvard University, 7 Divinity Avenue, Cambridge, MA 02138, USA. [4]Department of Chemistry and Chemical Biology, Harvard University, 12 Oxford Street, Cambridge, MA 02138, USA. [5]Department of Molecular and Cellular Biology, Harvard University, 52 Oxford Street, Cambridge, MA 02138, USA. [6]The Koch Institute for Integrative Cancer Research, Massachusetts Institute of Technology, 500 Main Street, Cambridge, MA 02142, USA. [7]Center for Environmental Health Sciences, Massachusetts Institute of Technology, 77 Massachusetts Avenue, Cambridge, MA 02139, USA. [8]Broad Institute of MIT and Harvard, 415 Main Street, Cambridge, MA 02142, USA. [9]Present address: Department of Chemistry, National Cheng Kung University, No.1, University Road, Tainan City 701, Taiwan. [10]Present address: Relay Therapeutics, Inc., 399 Binney Street, 2nd Floor, Cambridge, MA 02139, USA. [11]These authors contributed equally: Alex J. Callahan, Satish Gandhesiri. ✉e-mail: jmcgee@fogpharma.com; blp@mit.edu

cycles, but stability is constantly monitored throughout development[6]. Mirror-image D-peptides, on the other hand, are not recognized by natural proteases, and therefore display long half-lives in biological systems on their own[7,8]. As such, D-peptides are a privileged scaffold for peptide drug discovery, especially when combined with macrocyclic linkages, also termed staples. Despite these benefits, modern discovery efforts rarely utilize mirror-image peptides, as the high-throughput biological display techniques critical to modern peptide drug discovery efforts are not compatible with D-peptides. Development of mirror-image peptide-based therapeutics is limited to techniques that as of yet have not had sufficient throughput for modern drug discovery efforts.

Mirror-image phage display (MIPD) can generate high affinity D-peptide ligands to challenging protein interfaces, but its scope has remained limited. In MIPD, a library of L-peptides displayed on phage surfaces are screened against a 'bait' mirror-image protein[9]. By symmetry, the interaction of a D-peptide with an L-protein is identical to that of the screened L-peptide to a D-protein. As a result, the discovered L-peptide binders provide a blueprint for mirror-image D-peptide binders to the natural L-protein. This strategy has been employed to generate D-peptide ligands to therapeutically relevant protein targets, including mouse double minute 2 homolog (MDM2)[10], epidermal growth factor (EGF)[11], and amyloid beta (Aβ)[12]. Despite these successes, existing MIPD pipelines are low-throughput and, as a result, are not practical for modern drug discovery; to the best of our knowledge, MIPD has only been applied to nine protein targets[9–20] during the 30 years since its introduction.

A significant limitation to the broad application of MIPD relates to the challenges associated with accessing the required synthetic D-proteins[21]. Modern chemical protein synthesis (CPS) techniques have enabled access to a diverse range of synthetic proteins[22,23], but require individualized optimization for each target. Sequence-dependent synthetic challenges are ubiquitous, and chemical approaches to mitigate these problems are required. For example, aggregation disruptors[24–26], solubilizing tags[27,28], and ligation site alterations[29] are commonly used in modern CPS efforts. Our group implemented automated fast-flow peptide synthesis[30,31] (AFPS), an efficient technology which may obviate the need for individualized synthesis optimization. To our knowledge, 36 D-protein targets have been reported to date[2–4,11,14,15,19,20,29,32–45]. Despite the success of these strategies, their implementation requires many rounds of trial and error to address the specific complications of each sequence. As a result, each protein target presents a unique synthetic challenge that, although surmountable, requires significant effort. These considerations have rendered MIPD low throughput and, as a result, unable to compete with alternative discovery pipelines. A recently reported screening platform based on phage display generated high-affinity and conformationally constrained α-helical peptide binders[46].

Here, we use AFPS to prepare a panel of 12 diverse proteins (Fig. 1). We selected intracellular protein targets with underexplored biology (ERG and IRAK2), targets that are historically important to chemical protein synthesis (MDM2 and Barnase), or proteins that are emerging (CHIP, YAP1, NEMO, NEMO_iZIP) or classical (Myc-Max, Max-Max, BCL11a, and FKBP12) therapeutic targets. Of these sequences, eight had not been previously synthesized by chemical methods. We successfully isolate milligram amounts of these 12 proteins in both L- and D-forms (24 synthetic proteins total) (Fig. 1). We discover D-macrocyclic binders to two of the proteins synthesized here, the oncoprotein MDM2 and the C-terminus of Hsc70-interacting protein (CHIP). Analysis of the X-ray co-crystal structures between the discovered D-peptide binders and their protein targets reveals similar side-chain interactions to known binders that utilize different structural scaffolds, including loops and α-helices with opposite handedness. Together, these results represent a major advance in the throughput and general accessibility of D-peptide binder discovery.

## Results

### Synthesis of enantiomeric protein pairs with AFPS

We first defined a panel of 12 structurally diverse single-domain proteins for generation of synthetic L- and D-versions using AFPS

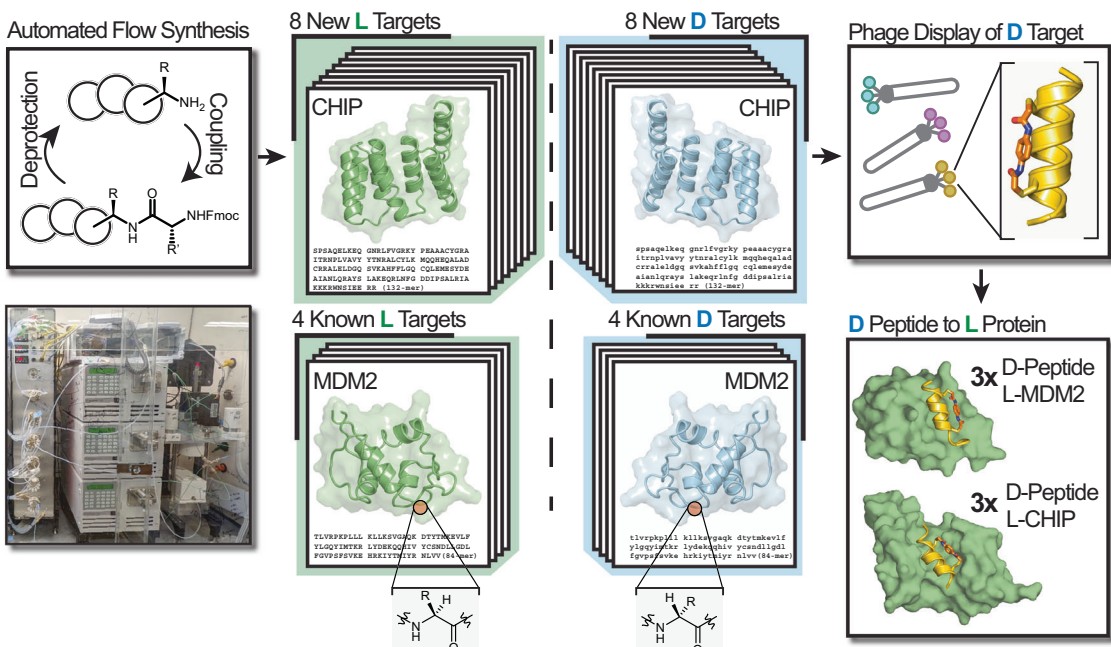

**Fig. 1 | Rapid generation of D-peptide binders leverages automated flow protein synthesis coupled with mirror-image phage display.** Automated flow protein synthesis can rapidly manufacture proteins with L- or D-amino acids. Subsequent folding and biochemical purification of each polypeptide chain affords synthetic protein pairs which are mirror images of each other. Phage display screening of the D-protein enantiomer with stapled alpha-helical peptides can reveal sequences with low-micromolar binding affinity. Mirroring of the hit peptides from L- to D-chirality affords binders to the native L-protein with similar affinity. Abbreviations: MDM2, mouse double minute 2 homolog; CHIP, C-terminus of Hsc70-interacting protein.

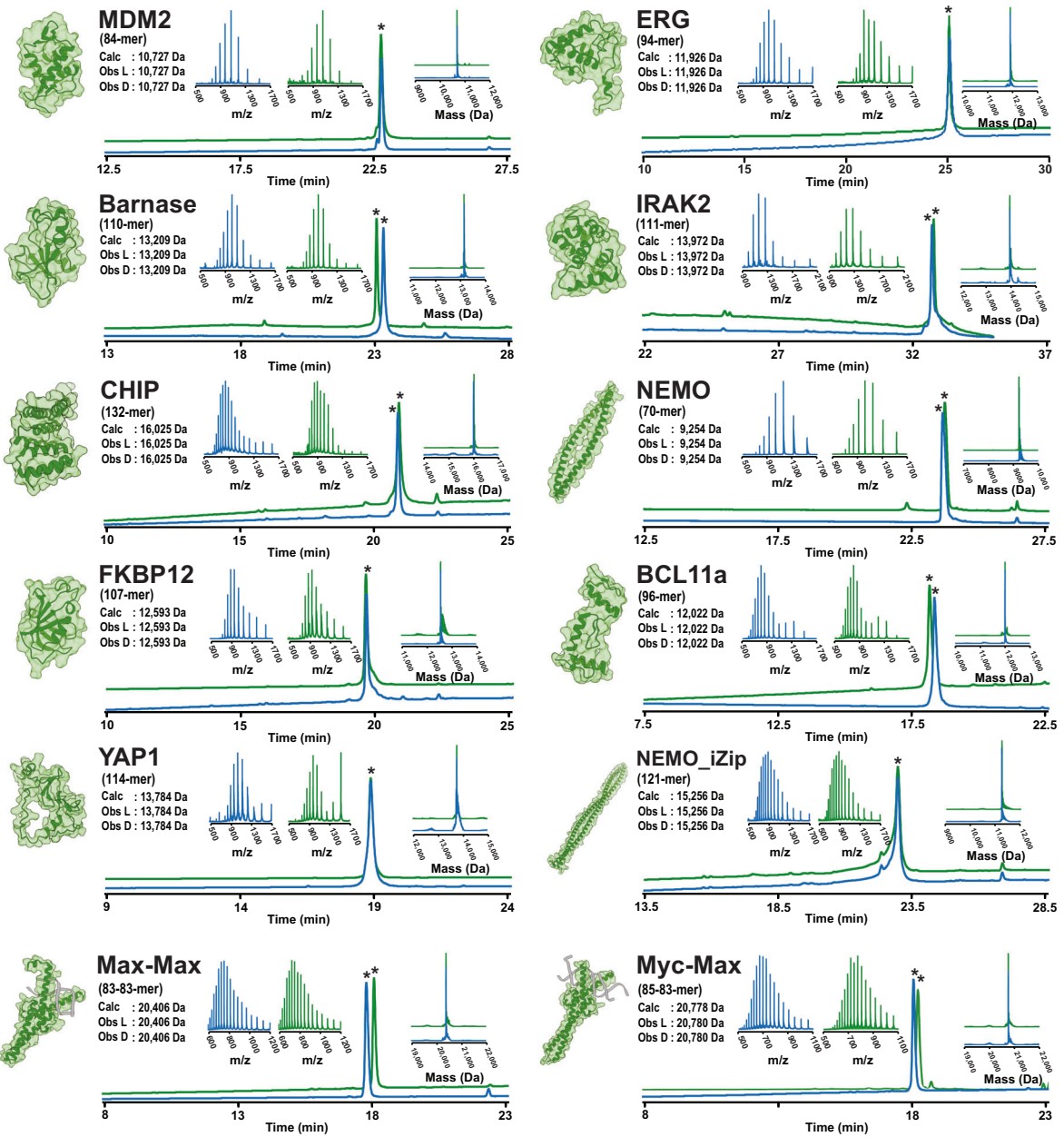

**Fig. 2 | Automated flow synthesis delivers diverse single-domain protein chains in both enantiomeric forms.** Analytical characterization of reverse phase-purified L- and D-chains from AFPS. Green traces show characterization of L-proteins, and blue traces show characterization of D-proteins. For each protein target, the following data are shown: (1) analytical HPLC trace of purified material recorded at 214 nm (bottom overlayed chromatograms); (2) integrated mass-to-charge spectrum of the total ion current (TIC) post-injection on a Q-TOF LC-MS instrument (separate insets on the left); and (3) the deconvolution of the TIC traces of (2) shown as overlayed insets on the right. Observed masses (abbreviated as Obs) from the deconvolution are shown along with the predicted values (abbreviated as Calc). The asterisks (*) in the analytical HPLC chromatograms indicate the major product peak corresponding to the integrated mass-to-charge and deconvoluted mass spectra depicted in the insets. The HPLC chromatograms for the L- and D-variants of barnase, BCL11a, Max-Max and Myc-Max dimers were acquired on different days, therefore variations in retention time were observed as a result of column conditioning. Previously reported structures are depicted for each of the targets, sourced from prior X-ray crystal structures available in the Protein Data Bank (PDB), NMR structures, or Alphafold predictions[50,51] (MDM2: 3FDO, ERG: 1SXE, Barnase: 1A2P, IRAK2: 3MOP, CHIP: 4KBQ, NEMO: 3BRV, FKBP12: 2PPN, BCL11a: 6KI6, YAP1: Alphafold prediction of Uniprot ID P46937 region 63–276, NEMO_iZIP: 6MI3, Myc-Max: 1HLO, Max-Max: 1NKP).

(Fig. 2). Two of these proteins, MDM2[10] and barnase[47], were previously prepared in their all-D forms (see SI Sections 2.3 and 2.5), and therefore can be used to assess the reliability and fidelity of the AFPS approach. We also chose two proteins that had previously only been synthesized in their native L-forms: Myc-Max, and Max-Max[48] (see SI Sections 2.13–2.19). We chose an additional eight single-domain proteins ranging in length from 70–132 amino acids with no reported chemical synthesis: ERG (PNT domain) (see SI Section 2.4),

IRAK2 (death domain) (see SI Section 2.6), CHIP (tetranucleotide repeat domain) (see SI Section 2.7), NEMO (coiled coil domain) (see SI Section 2.8), FKBP12 (see SI Section 2.9), BCL11a (Zn finger domains) (see SI Section 2.10), YAP1 (ww1-ww2 domains) (see SI Section 2.11) and NEMO_iZIP (coiled coil domain with iZip adapters[49]) (see SI Section 2.12). In all cases, we appended a biotin unit at the N-terminus of the protein through a PEG$_{12}$ linker to facilitate phage display screening.

We used AFPS to prepare each polypeptide chain in a stepwise fashion, with total synthesis times ranging from 4–7 h, using three separate AFPS instruments. After we released the proteins from the H-Rink Amide solid support, we isolated them by precipitation with cold diethyl ether (see Methods). We used analytical reverse phase HPLC (RP-HPLC) and liquid chromatography–mass spectrometry (LC-MS) to analyze crude peptide mixtures and preparative mass-directed RP-HPLC to purify them. We typically obtained ~5 μmol of crude polypeptide powder from individual synthesis runs, and purifying ~3 μmol of this material afforded ~0.3 μmol of pure peptide. Gradient shape for the preparative purification was determined by a preliminary run at low protein loading on the same column used for purification (see SI Section 1.4). Typical yields are on the order of 1–10 mg (0.07–0.7 μmol, 0.3%–3% isolated yield based on resin loading). In all instances, a single synthesis experiment afforded sufficient material after folding for phage selections and biochemical validation.

### Folding of diverse D-proteins with preparative size exclusion chromatography

Using a generalized folding protocol, each synthetic protein was folded to a homogeneous product as analyzed by size-exclusion chromatography (SEC). We set out to identify appropriate folding conditions for our 12 enantiomeric protein pairs and began by adopting the protocols that had been used to successfully isolate folded protein in an aqueous solution for the few targets for which literature reports exist. Specifically, L/D Barnase[31] and L/D MDM2[52] were previously isolated via stepwise dilution from a high concentration of guanidine hydrochloride. Subsequent purification using semi-preparative SEC achieves yields ranging from 4.5 to 10.6 nmol for all four targets. Here, using this technique, we were unable to identify any dilution conditions under which we could isolate folded BCL11a or IRAK2. Instead, we observed significant precipitation for both target pairs, with any remaining solubilized material eluting in the exclusion volume of the column, indicating an apparent molecular weight at least 10-fold over the expected value, consistent with the formation of soluble aggregates.

To address these complications, we adapted a refolding technique utilizing SEC to isolate the remaining protein targets[53,54]. Purified proteins were dissolved in a denaturing buffer, typically a 10 mg/mL solution containing 6 M guanidine hydrochloride buffered to pH 7.5 with 50 mM tris(hydroxymethyl)aminomethane hydrochloride (TRIS HCl) and including 50 mM dithiothreitol (DTT). The mixture was then submitted to semi-preparative SEC coupled to an HPLC instrument. The denaturant is removed as the protein progresses through the SEC column, and the resulting folded protein can be separated from off-target aggregates based on their different retention times. The running buffer of the SEC could be tuned to the requirements of each protein and unless otherwise specified was 50 mM HEPES, 150 mM NaCl, 0.5 mM DTT, 5% glycerol (v/v), at pH 7.5. DTT was omitted in cases where the protein contained no free sulfhydryl groups (NEMO, NEMO_iZIP, Myc-Max, and Max-Max). For BCL11a, which contains three Zn-finger motifs, both the denaturing buffer and the running buffer were adjusted to contain $ZnCl_2$ with additional alterations to prevent precipitation of Zn salts. In each case, the folding protocol afforded material as major elution peaks with retention times consistent with the calculated molecular weights (SI Sections 2.20 to 2.31). This set includes BCL11a, and IRAK2 for which we were unable to isolate folded material by any other technique. Some proteins elute as peaks with apparent molecular weight larger than would be otherwise expected, including CHIP, NEMO, NEMO_iZIP, and BCL11a. In the case of NEMO, and NEMO_iZIP, this behavior has been established and has been attributed to their extended structures[49,55].

### Synthetic D- and L-proteins display expected biochemical activity

For each enantiomeric pair generated using AFPS, we used biochemical assays to confirm that the activities of the synthetic proteins are similar to their recombinant versions. For proteins with known binding partners, we measured the affinities of the synthetic proteins to fluorophore-tagged probes of the appropriate chirality using time-resolved fluorescence resonance energy transfer (TR-FRET). Binder substrates were modified with 5(6)-carboxyfluorescein (FAM), and the biotinylated target proteins were complexed with terbium-labeled streptavidin. Binding was measured as an increase in fluorescence quenching between the two fluorophores. For BCL11a binding to DNA (Fig. 3A and Supplementary Table 1), CHIP binding to HSP peptide (Fig. 3B and Supplementary Table 2), MDM2 binding to a p53-derived peptide (Fig. 3I and Supplementary Table 3), Max-Max binding to E-Box DNA (Fig. 3E and Supplementary Table 5), and Myc-Max binding to E-Box DNA (Fig. 3F and Supplementary Table 4), each synthetic chiral protein displayed apparent dissociation constants ($K_D$) for the L- and D-proteins consistent with those of the recombinant proteins (Fig. 3L). We next used surface plasmon resonance (SPR) binding assays with substrates of the matching chirality to assess binding of NEMO to iKKb peptide (see Supplementary Fig. 2), NEMO_iZip binding to iKKb peptide (see Supplementary Fig. 5), YAP1 binding to dendrin (see Supplementary Fig. 4), and L-FKBP12 binding to rapamycin followed by mTOR (see Supplementary Fig. 3). Again, in all instances, the synthetic proteins displayed binding affinities similar to the recombinantly derived proteins, suggesting that the synthetic proteins derived from AFPS are folding into bioactive tertiary structures.

Synthetic L/D barnase from AFPS displays ribonucleolytic activity that is selective for the chirality of its nucleotide substrate. Barnase catalytically cleaves a fluorescently quenched reporter RNA substrate[57] composed of a tetranucleotide flanked with FAM and 6-carboxytetramethylrhodamine (6-TAMRA) labels on either end (FAM-DdA-DrU-DdA-DdA-TAMRA), where each nucleotide is specified as the L- or D-enantiomer and the ribose (r) or deoxyribose (d) sugar. We tracked the barnase-mediated cleavage of the substrate by assessing the FRET between the FAM and TAMRA fluorophores, which is abolished upon cleavage. When the chirality of the substrate and protein were matched, as for L-barnase to the D-substrate FAM-DdA-DrU-DdA-DdA-TAMRA, and D-barnase to the L-substrate FAM-LdA-LrU-LdA-LdA-TAMRA, we observed rapid hydrolysis with extracted kinetic constants that are similar to previously reported values (Fig. 3J, K and Supplementary Table 6)[47]. In the mismatched cases, L-barnase to L-substrate and D-barnase to D-substrate, we observed no catalysis on the timescale of our observation (Fig. 3J, K and Supplementary Fig. 1). These results reconfirm the structural and biological integrity of the folded synthetic proteins, indicating that their enantiomeric purities are retained throughout the synthesis and purification processes.

For protein targets with no known binding partners and no available activity assays (ERG and IRAK2), we recorded circular dichroism (CD) spectra and compared them to recombinantly derived material. Both targets are reported to be mostly alpha-helical[58,59], and we confirmed this feature for the recombinant proteins. The CD spectra for the synthetically derived L-proteins closely matched those of the recombinant versions, indicating that the synthetic material is forming secondary structures in similar proportions (Fig. 3C, G and SI Section 1.12). Both D-ERG and D-IRAK2 display a CD signal with similar absolute intensities as the corresponding synthetic L- and recombinant proteins, but with inverted sign, consistent with the formation of mirror-image secondary structures. Furthermore, we found that both synthetic protein pairs form folded structures with melting temperatures ($T_m$) close to those of the recombinantly derived proteins, as shown by tracking the intensity of the α-helical signature as a function of temperature (Fig. 3D, H and SI Section 1.12). Together, these data suggest that the synthetically derived L/D-ERG and L/D-IRAK2 form tertiary structures that closely resemble the native proteins.

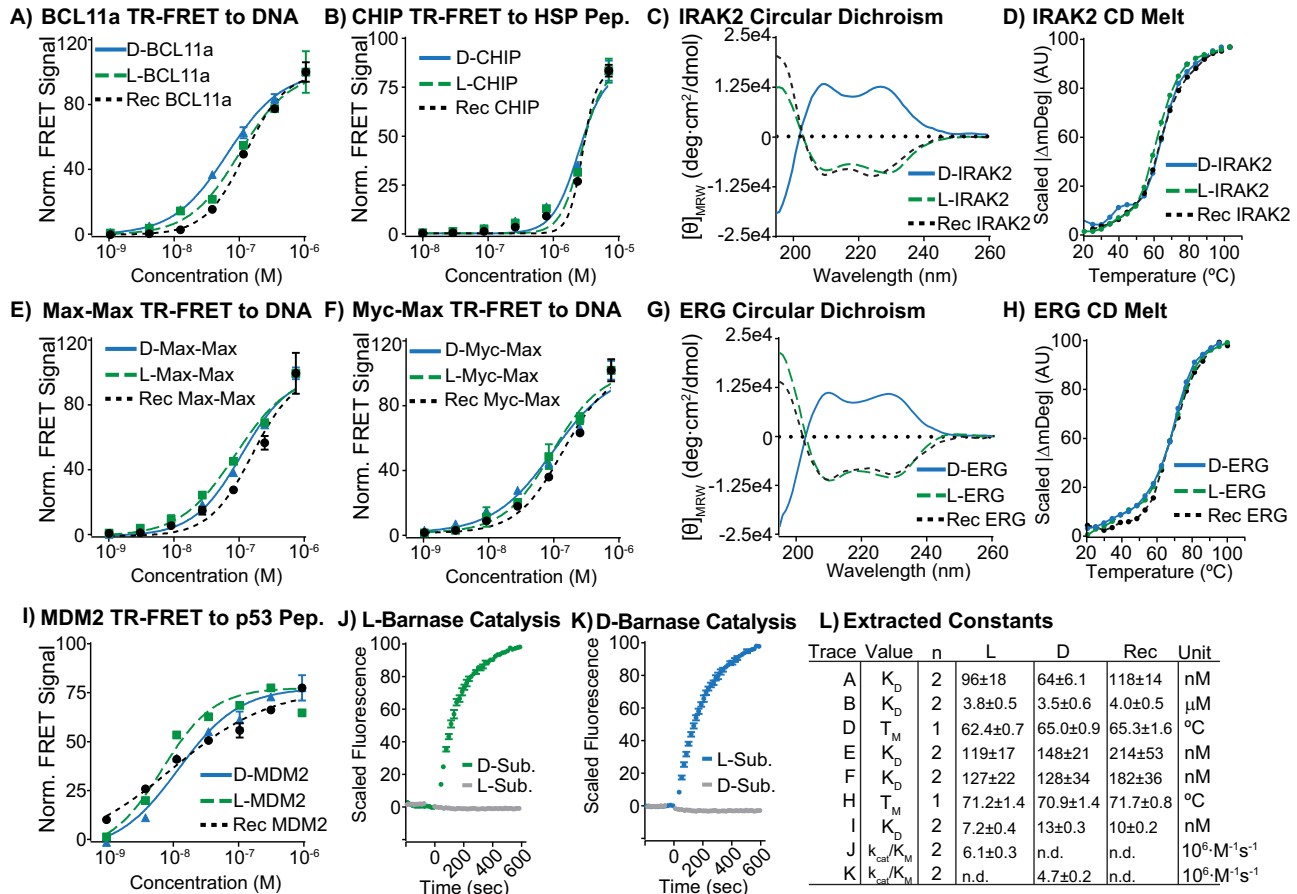

**Fig. 3 | Synthetic D-proteins display similar bioactivity to their synthetic L- and recombinant counterparts.** Binding data for five synthetic protein targets to peptide ligands of the appropriate chirality are shown. Experimental details are outlined in the Supplementary Information Sections 1.7–1.12. **A** Synthetic D, L and recombinant BCL11a bind a model DNA oligonucleotide of the γ-globulin promoter[56] measured as an increase in FRET efficiency between FAM and terbium streptavidin. The same method was used to measure binding for the studies in (**B**, **E**, **F**, **I**). **B** Synthetic D, L and recombinant CHIP bind a peptide model of the heat shock protein 70 (Hsp70) C-terminus. **C** Synthetic D, L and recombinant IRAK2 display similar proportions of secondary structure by circular dichroism (CD) spectra recorded from 195 to 260 nm at 0.1 mg/mL. **D** Synthetic L, D and recombinant IRAK2 have similar melting temperatures ($T_m$) determined by variable temperature CD monitored at 222 nm from 20 °C to 100 °C in 5 °C steps. **E** Synthetic D, L and recombinant Max-Max bind a model DNA oligonucleotide of E-box DNA[47]. **F** Synthetic D, L and recombinant Myc-Max bind a model DNA oligonucleotide of

E-box DNA[48]. **G** Synthetic D, L and recombinant ERG display similar proportions of secondary structure by CD spectra recorded as described in (**C**). **H** Synthetic L, D and recombinant ERG have similar melting temperatures determined as described in (**D**). **I** Synthetic D, L and recombinant MDM2 bind a p53-derived model peptide. For studies described in (**A**, **B**, **E**, **F**, **I**), the chirality of the binder was adjusted to that of the protein. **J**, **K** Synthetic L- and D-barnase selectively catalyze the hydrolysis of a stereochemically matched RNA substrate. Catalytic activity was measured as an increase in fluorescence intensity (see SI Section 1.11). No cleavage for the mismatched substrate pairs (L-RNA to L-barnase and D-RNA to D-barnase) was observed. The data in (**A**, **B**, **E**, **F**, **I**, **J**, **K**) are presented as mean ± SD, $n = 2$ technical replicates. **L** Experimentally determined biophysical and biochemical parameters. The reported values are based on the data presented in Supplementary Tables 1–6 and are obtained by applying the corresponding fitting models described in the Supplementary Information Sections 1.7–1.12 (n.d. = not determined).

## Generation of macrocyclic D-peptide ligands to MDM2 using mirror-image phage display

Having validated the structure and function of our synthetic D-proteins, we sought to benchmark our MIPD screening platform with a model target, MDM2, an E3 ubiquitin ligase that recognizes the FXXXWXXL motif present on its substrate, p53, a critical mediator of cell cycle arrest, senescence and apoptosis[60]. As a result, p53 degradation driven by upregulation of MDM2 is found to accelerate growth in a variety of cancers[61]. To disrupt p53 recognition, and thus cancer progression, a number of high-affinity MDM2 binders have been developed and evaluated in the clinic[62]. In particular, highly-potent biologically stable linear D-peptide ligands to MDM2 have been generated with mirror-image phage display, including $^{D}$PMI-α ($K_D$ = 219 nM)[63], $^{D}$PMI-β ($K_D$ = 219 nM)[10], and $^{D}$PMI-γ ($K_D$ = 53 nM)[63]. We set out to investigate if our MIPD platform could rediscover similar motifs, and perhaps generate new candidates for MDM2 inhibition.

To that end, we successfully isolated synthetic mirror image MDM2, and carried it into our MIPD platform[46].

We identified three binding clusters selective for D-MDM2 with phage display screening (Fig. 4A and SI Section 1.26). In brief, immobilized D-MDM2 from AFPS was screened against an unbiased phage library of $10^8$ members displaying macrocyclic α-helical peptides. We sequenced phage particles that remained bound to the targets after washing using next-generation sequencing. Comparisons of sequencing reads to a spiked-in internal reference allowed for a semi-quantitative read-out of binding affinity as a function of protein bait concentration. The resulting hit sequences that displayed dose-response were clustered into binder families using hierarchical statistical clustering. We identified three such clusters to D-MDM2, and logo plots of their sequence overlaps are shown in Fig. 4A (MDM2.C1-MDM2.C3). Consistent with previously described α-helical peptide binders to MDM2[64,65], all three clusters contain anchoring N-terminal

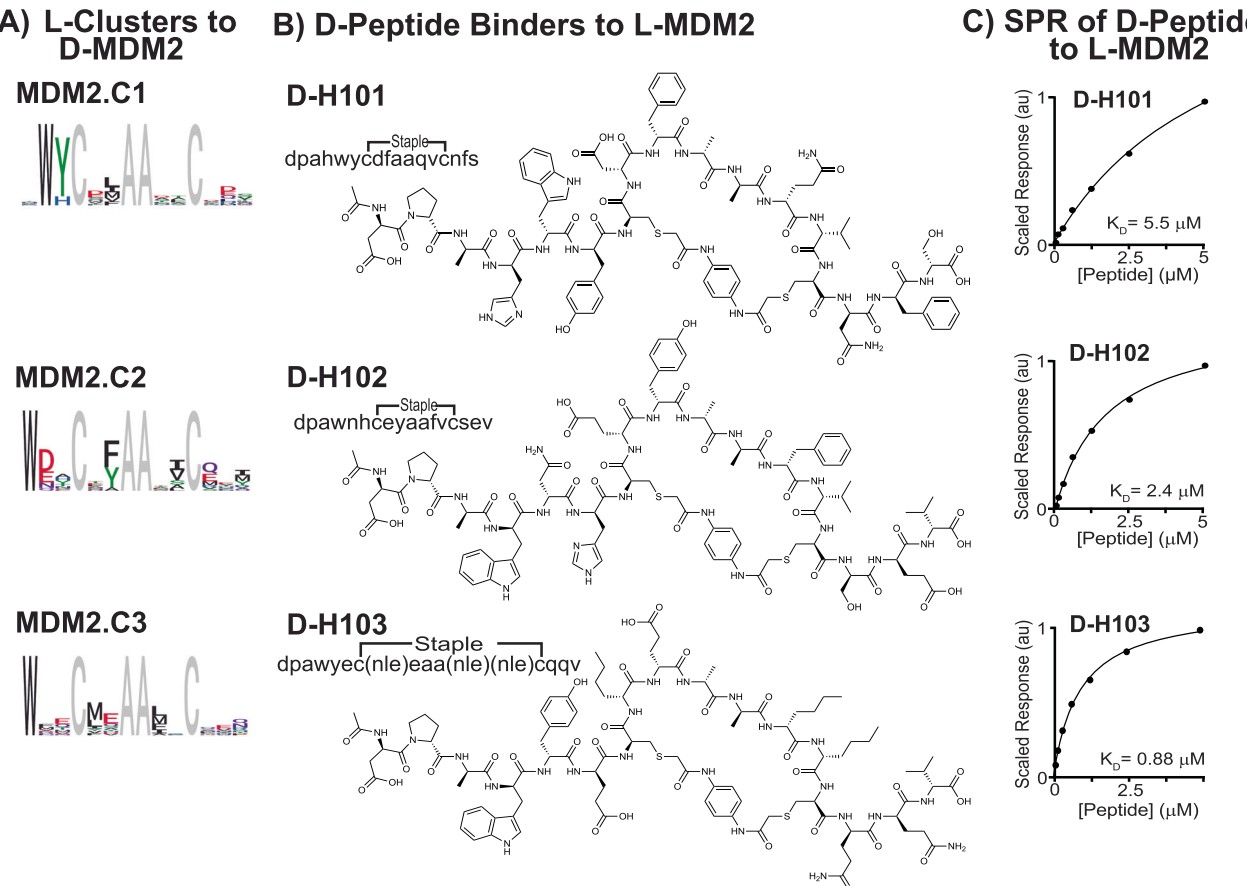

**A) L-Clusters to D-MDM2**

MDM2.C1

MDM2.C2

MDM2.C3

**B) D-Peptide Binders to L-MDM2**

D-H101
dpahwycdfaaqvcnfs — Staple

D-H102
dpawnhceyaafvcsev — Staple

D-H103
dpawyec(nle)eaa(nle)(nle)cqqv — Staple

**C) SPR of D-Peptides to L-MDM2**

D-H101  $K_D$ = 5.5 μM

D-H102  $K_D$ = 2.4 μM

D-H103  $K_D$ = 0.88 μM

**Fig. 4 | Mirror-image phage display of D-MDM2 generates macrocyclic D-peptide binders to L-MDM2. A** Logo plots of the three L-peptide clusters that bind to D-MDM2 are shown with fixed residues in gray. **B** Identities and chemical structures of selected D-macrocycles selected for individual synthesis are shown. The selected peptide sequences are mirror-image members of the cluster shown to their left. The individual sequences chosen for validation were synthesized with D-norleucine instead of D-methionine to avoid possible side-chain oxidation reactions. Lower-case letters in the sequences denote single-letter abbreviations for D-amino acids, nle = D-norleucine. **C** The discovered macrocyclic D-peptides bind recombinant MDM2. Steady-state SPR sensorgrams of the D-binders to D-MDM2 are shown with extracted binding affinities (see Supplementary Figs. 6–8).

tryptophan residues (dW2 in MDM2.C1, dW1 in MDM2.C2, and dW1 in MDM2.C3), and show conserved hydrophobic residues on the same α-helical face as the tryptophan residue ($i,i$ + 4/5 and $i,i$ + 8/9 relative to dW on each cluster). To validate the specificity of each binding cluster, we synthesized representative binding peptides in the D-form (Fig. 4B, D-H101 from MDM2.C1, D-H102 from MDM2.C2, and D-H103 from MDM2.C3) and measured their affinity to recombinant MDM2 with SPR. We observed binding constants of 5.5 μM, 2.4 μM, and 0.88 μM, respectively (Fig. 4C and Supplementary Figs. 6–8).

To further investigate the binding modes of each binding cluster, we solved the X-ray co-crystal structures of D-H101 (Fig. 5A, resolution 1.64 Å), D-H102 (Fig. 5B, resolution 1.30 Å), and D-H103 (Figs. 5C, resolution 1.90 Å) with recombinant MDM2 (see SI Section 3). All three peptides form left-handed α-helices that engage the same hydrophobic groove on MDM2 but make use of different side-chain interactions. D-H101 and D-H102 use similar amino acids to interact with MDM2 at its N-terminus (dW5/dW4, dY6/dH6, and dF9/dY9), but diverge towards the C-terminus where the side-chain conformations are altered (dQ12/dF12 and dE16/dF16) (Fig. 5A, B). Beyond the conserved N-terminal aromatic residues, D-H103 makes use of altered side-chains to interact with MDM2 compared to either D-H101 or D-H102 (Fig. 5C vs Fig. 5A, B).

Consistent with their sequence similarities, D-H101 utilizes similar side-chain interactions to two previously reported α-helical peptide binders to MDM2. Overlay of the known D-peptide MDM2 binder dPMI[1–5, 9–12][64] onto the D-H101 MDM2 co-crystal structure revealed that both peptides present into the same groove on MDM2 with the same axial direction (N to C), and make use of similar side-chains (Fig. 5D). Chemically related side-chains project with similar orientations for both peptides, perhaps due to their matched left-handed helices. Matching helicity, however, is not an exclusive requirement to make use of similar side-chain interactions. ATSP-7041 is a high-affinity L-peptide binder to MDM2 ($K_D$ = 0.9 nM)[65] and its structure overlaid onto the D-H101 MDM2 co-crystal structure is shown in Fig. 5C. Despite ATSP-7041 forming a right-handed alpha helix, both peptides appear to use similar binding modes. Important side-chain interactions of ATSP-7041 closely overlap with equivalent interactions for D-H101 (Fig. 5C, dW5/F3, dY6/Y6, dF9/W7, and dV13/Cba10).

### Generation of macrocyclic D-peptide ligands to CHIP using mirror-image phage display

Not limited to known D-proteins, we nominated an original target for MIPD screening, CHIP, and identified three macrocyclic D-peptide binders. CHIP is a member of the RING/U-Box family of E3 ligases that direct the ubiquitination of chaperones[66] and their bound client proteins[67,68] (Fig. 6A). This interaction is driven by a common (I/M) EEVD motif present on the C-terminus of heat shock protein 70 (Hsp70) and Hsp90 family members that is recognized by the TPR domain of CHIP[69] (Fig. 6B). Degradation driven by CHIP is active in a diverse range of cellular processes from monitoring protein quality control[68] to mediating interferon-γ signaling[70]. Masking of the CHIP-TPR domain can in theory modulate these signaling pathways and as a

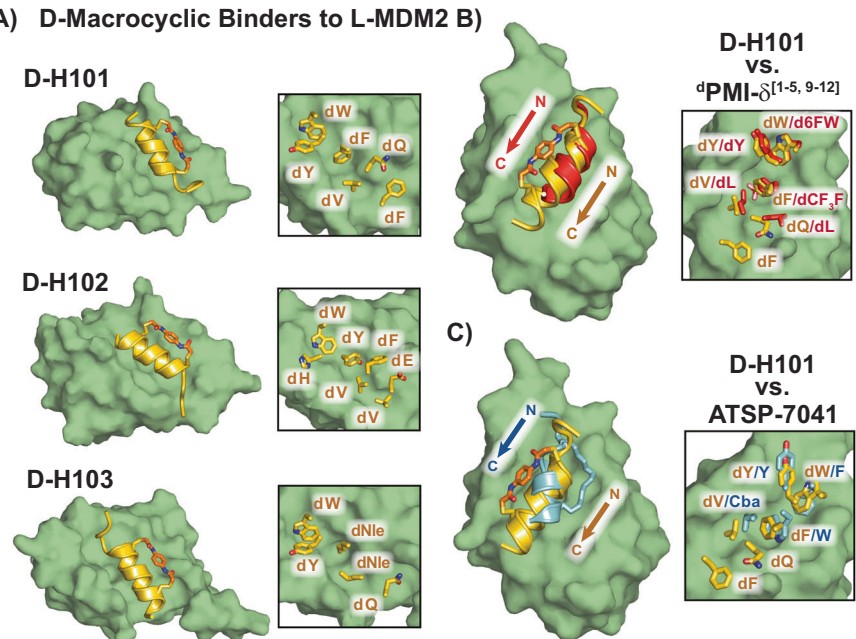

**Fig. 5 | Discovered D-macrocycles bind to a common hydrophobic groove on MDM2. A** X-ray co-crystal structures of macrocyclic D-peptides from each cluster to L-MDM2 are shown. Side-chains projecting into the MDM2 surface are highlighted. **B** The co-crystal structure of D-H101 with MDM2 is shown overlaid with the known D-peptide binder ᵈPMI-δ[1–5, 9–12][18,63] (PDB ID: 6KZU). Side-chains that project into the MDM2 surface are highlighted in the inset. The lowercase d inserted before a single letter amino acid abbreviation denotes D-amino acids. Nle = norleucine. **C** The co-crystal structure of D-H101 with MDM2 is shown overlaid with the known L-peptide binder and ATSP-7041[63] (PDB ID: 4N5T). Side-chains that project into the MDM2 surface are highlighted in the inset.

result is actively being pursued[69,71]. In addition, CHIP belongs to the same protein family as MDM2 (E3 ubiquitin ligases) and has potential therapeutic relevance by being implicated in neurological disease. For these reasons, we hypothesized that this domain would be a valuable target for D-peptide discovery, and successfully isolated the protein in its mirror image form for MIPD screening. From a single-round phage screen, we identified three clusters that displayed robust binding to CHIP (CHIP.C1–CHIP.C3) (SI Section 1.27). All three clusters feature conserved terminal residues with carboxylate functionalities (dD3 in CHIP.C1, dD1/dE2 in CHIP.C2, and dD12 in CHIP.C3), consistent with the conserved terminal aspartic acid residues found on CHIP substrates. Hydrophobic residues are retained in the $i, i+4/5$ positions relative to the anchoring aspartic acid residue: dW6 in CHIP.C1, dW6 in CHIP.C2, and dA8 in CHIP.C3, presumably for interaction with the hydrophobic TPR groove. We validated binding of selected sequences from the three clusters to the recombinant CHIP-TRP domain ($IC_{50}$ values of 0.47 μM, 0.55 μM, and 1.23 μM, respectively, Fig. 6C) in a competition fluorescence polarization assay against FAM-Hsp70 loaded CHIP (see Supplementary Fig. 15).

X-ray co-crystal structures of each discovered D-macrocycle with recombinant CHIP revealed a common binding pocket despite variable side-chain interactions. We solved the X-ray co-crystal structures of each peptide with recombinant CHIP (Fig. 6B–D, 1.72 Å, 1.59 Å, and 1.76 Å resolution, respectively) (see SI Section 3). As predicted, all three peptides form left-handed α-helices to engage the same groove, though D-H202 binds in the reverse direction compared to D-201 and D-203 (Fig. 6C). We also solved the co-crystal structure of CHIP in complex with D-H204 at a resolution of 2.21 Å. We identified D-H203 and D-H204 from the same phage cluster. As expected, they share a similar amino acid sequence and binding mode. Despite varying residues, side-chains projecting into the CHIP surface across all binders have significant spatial overlap (Fig. 6D). Two loci for hydrophobic interactions on the CHIP surface in particular show strong side-chain overlap: dW7/dL6/dW7, and dF8/Staple/dM4 (of D-H201, D-H202, and D-H203, respectively). Additionally, the common terminal carboxylate-

containing residues engage in similar hydrogen bonding networks with CHIP. These similarities extend to known biological ligands to CHIP, and an overlay of a co-crystal structure of the C-terminus of Hsc70 onto D-H201 is shown in Fig. 6E. Biological ligands to the CHIP TPR domain are known to form distinct kinked backbone conformations[69], unlike the left-handed alpha-helices of the discovered macrocycles. Despite this feature, the biological ligand utilizes similar side-chain contacts to engage CHIP. Both hydrophobic side-chain sites found in the discovered macrocycles (dF8/V4 and dW7/I1 in D-H201 and Hsc70, respectively) and the terminal carboxylate are also found in the biological CHIP ligand. These similarities indicate that the discovered macrocyclic D-peptides utilize a native binding mode translated into a left-handed alpha-helical interface not characterized previously.

## Discussion

Peptide scaffolds are increasingly more often applied to new therapeutic targets and D-peptides in particular have great potential to be broadly applicable. Native, canonical peptides are notoriously unstable in vivo, and this reputation has complicated their application as therapeutics[72]. Increasingly effective and easily accessible chemical stabilization strategies have revitalized peptides as effective therapeutic scaffolds[6]. With judicious optimization, newly reported peptide therapeutics routinely display long biological half-lives. For example, after optimization, the candidate PCSK9 inhibitor MK-0616 displays nearly complete stability against key biological proteases[73], and as a result is dosed orally in its Phase 2b clinical trial[74]. However, such achievements still require a complex and meticulous drug development process. Mirror-image D-peptides, on the other hand, should display substantial metabolic stabilities. Routine implementation of screening strategies that afford D-peptide ligands stands to greatly simplify and accelerate peptide drug development.

We demonstrate a robust and scalable platform that uses automated flow synthesis technology to facilitate access to mirror-image proteins for D-peptide discovery using mirror-image phage display.

**Fig. 6 | D-macrocyclic peptides identified from MIPD are high-affinity ligands to CHIP. A** The E3 ubiquitin ligase CHIP marks chaperone bound client proteins for degradation through recognition of a C-terminal peptide motif (PDB ID: 4KBQ). **B** The TPR domain of CHIP recognizes a C-terminal IEEVD motif. **C** Mirror-image phage display of the TPR domain of CHIP affords three D-macrocyclic peptide binders to a common groove. X-ray co-crystal structures of the identified sequences are shown projected onto the CHIP surface from D-H202, with annotated helical directions. D-H201 is shown in yellow, D-H202 in blue, and D-H203 in purple. Lower-case letters in the sequences denote single-letter abbreviations for D-amino acids. Binding affinities were determined from competition FP of each peptide to FAM-Hsp70-peptide bound recombinant CHIP. **D** All three binders utilize different chemical contacts to engage the same pockets on CHIP, despite different helical directions. Interacting side-chains from the X-ray co-crystal structures of each binder are shown overlaid. Side-chain colorings are as in (**C**). **E** The discovered binders utilize unique side-chains to engage contact sites of the native peptide substrate. The X-ray co-crystal structure of D-H201 is shown overlaid with the Hsp70 recognition peptide (PDB ID: 4KBQ). Overlapping interactions are highlighted, with D-H201 in yellow and Hsp70 peptide in pink. The lower-case d inserted before a single letter amino acid abbreviation denotes D-amino acids.

Among the main obstacles to the broad implementation of MIPD are the inconsistent and unpredictable outcomes of the total chemical synthesis of the requisite mirror-image proteins and their fragments. In this work, we demonstrate that AFPS technology can reliably supply mirror-image proteins for MIPD in a single shot. We successfully isolated 12 mirror-image proteins with AFPS, 8 of which have no previously reported syntheses. Without AFPS, these efforts would likely require significant individualized synthetic route optimization and method troubleshooting. We acknowledge that individual optimization may be needed for the flow synthesis of specific protein sequences, identified in our laboratory on a case-by-case basis. This was not the case here, however, as each target protein was synthesized successfully on the initial attempts according to our established protocol.

In our work, these mirror-image proteins were rapidly prepared in a single shot for screening in a standardized format and the ease of synthesis benefitted from their relatively small size (<150 amino acids). Making MIPD routinely applicable to larger protein targets may likely require combining flow synthesis with ligation methods. Our ongoing efforts are focused on leveraging these two powerful synthetic approaches. Continued improvement of D-protein synthesis quality is key to making MIPD practical for modern screening pipelines.

With practical and reliable access to mirror-image proteins, we describe the discovery of six macrocyclic D-peptide ligands to two protein targets. We first validated our discovery platform with screening of MDM2, a negative regulator of p53 known to be overexpressed in a variety of cancers[61]. MDM2 is one of a few previously reported targets for mirror-image phage display[10], and we successfully rediscovered sequence motifs that are similar to these earlier reports. Furthermore, we recorded X-ray co-crystal structures of three mirror image binders to MDM2. Collectively, these results confirm the fidelity of our synthesis approach, and in addition provide new therapeutic leads for targeting MDM2. We further nominated CHIP for mirror image phage display screening, and again identified three macrocyclic D-peptide ligands. CHIP is an E3 ubiquitin ligase that directs the

degradation of cellular chaperones and their bound client proteins[66]. Inhibition of the CHIP ligand binding site is an emerging approach to modulate interferon-γ signaling[70]. We anticipated that mirror-image peptide ligands could prove a valuable tool to engage this site. The three discovered peptide ligands all bound to recombinant CHIP and were able to displace a native CHIP ligand at low micromolar concentrations. Characterization of each mirror-image ligand by X-ray co-crystal structures with recombinant CHIP revealed a similar binding site to the native CHIP ligands, indicating their potential to disrupt this interaction.

Collectively, the results reported here represent a significant fraction of the successful D-protein syntheses to date. Widespread application of MIPD is hindered by the individualized attention required to prepare the mirror-image D-protein substrates, and we show that APFS addresses these challenges by delivering mirror-image proteins for screening using a standardized, rapid format. We anticipate that MIPD enabled by AFPS is poised to revitalize campaigns to generate mirror-image binders to existing and emerging protein targets.

## Methods

### General methods

Unless otherwise specified, all reactions with polypeptides, proteins, and protein oxidative addition complexes were set up on the bench top and carried out under ambient conditions. Unless stated otherwise, all small (≤1 mL) volumes were measured using Eppendorf Research® plus, single-channel, variable, mechanical pipettes (referred to as mechanical pipettes). Universal low retention pipet tips (10 μL, 200 μL and 1000 μL sizes) were purchased from VWR International (Philadelphia, PA). Care was taken to use the appropriate mechanical pipette/tip combinations to ensure a dispensing error of ≤ 2% for volumes between 10 μL and 1000 μL and ≤4% for volumes between 1 μL and 10 μL. To avoid loss of protein due to non-specific adsorption, plastic tubes (Eppendorf Protein LoBind® tubes, 0.5, 1.5, and 2.0 mL) were used in all cases after folding. The weight of lyophilized powders of the peptides was measured using analytical scales (XS205DU Analytical Balance, Mettler-Toledo) with an attached SPI Westek Workstation Still Air Ionizer. All fluorenylmethyloxycarbonyl (Fmoc)-protected D-amino acids were lyophilized for at least 12 h before use to remove trace solvents.

### Materials for peptide synthesis

All reagents were purchased and used as received unless otherwise noted. Fmoc-protected L-amino acids (Fmoc-Ala-OH•H₂O, Fmoc-Cys(Trt)-OH, Fmoc-Asp(OtBu)-OH, Fmoc-Glu(OtBu)-OH, Fmoc-Phe-OH, Fmoc-Gly-OH, Fmoc-Ile-OH, Fmoc-Lys(Boc)-OH, Fmoc-Leu-OH, Fmoc-Met-OH, Fmoc-Asn(Trt)-OH, Fmoc-Pro-OH, Fmoc-Gln(Trt)-OH, Fmoc-Arg(Pbf)-OH, Fmoc-Ser(tBu)-OH, Fmoc-Thr(tBu), Fmoc-Val-OH, Fmoc-Trp(Boc)-OH, and Fmoc-Tyr(OtBu)-OH) were purchased from the Novabiochem line (Millipore-Sigma). L-Fmoc-His(Boc)-OH was purchased from ChemPep. L-Fmoc-Nle-OH was purchased from Chem-Impex. Fmoc-protected D-amino acids (Fmoc-D-Ala-OH•H₂O, Fmoc-D-Cys(Trt)-OH, Fmoc-D-Asp(OtBu)-OH, Fmoc-D-Glu(OtBu)-OH, Fmoc-D-Phe-OH, Fmoc-D-His(Boc)-OH, Fmoc-D-Ile-OH, Fmoc-D-Lys(Boc)-OH, Fmoc-D-Leu-OH, Fmoc-D-Met-OH, Fmoc-D-Asn(Trt)-OH, Fmoc-D-Pro-OH, Fmoc-D-Gln(Trt)-OH, Fmoc-D-Arg(Pbf)-OH, Fmoc-D-Ser(tBu)-OH, Fmoc-D-Thr(tBu)-OH, Fmoc-D-Val-OH, Fmoc-D-Trp(Boc)-OH, and Fmoc-D-Tyr(OtBu)-OH) were purchased from both the Novabiochem line from Millipore-Sigma and Chem-Impex. Fmoc-D-Nle-OH was purchased from Chem-Impex. O-(7-azabenzotriazol-1-yl)-N,N,N′,N′-tetramethyluronium hexafluorophosphate (HATU, ≥97.0%) and (7-aza-benzotriazol-1-yloxy)trispyrrolidinophosphonium hexafluorophosphate (PyAOP, ≥97.0%) were purchased from P3 Biosystems. Biosynthesis OmniSolv® grade N,N-dimethylformamide (DMF) was purchased from EMD Millipore, and stored over AldraAmine trapping

agents (for 1000–4000 mL, Millipore-Sigma catalog number Z511706). Diisopropylethylamine (DIEA; 99.5%, biotech grade, catalog number 387649) was purchased from Sigma-Aldrich and purified by passage through an activated alumina column (Pure Process Technology solvent purification system). Piperidine (ACS reagent, ≥99.0%), trifluoroacetic acid (TFA, HPLC grade, ≥99.0%), triisopropylsilane (TIPS, ≥98.0%), acetonitrile (HPLC grade), formic acid (FA, ≥95.0%), phenol (ACS reagent, ≥99.0%), diethyl ether (Et₂O, ≥99.7%, containing 1 ppm BHT as inhibitor), and 1,2-ethanedithiol (EDT, GC grade, ≥98.0%) were purchased from Sigma-Aldrich. H-Rink Amide ChemMatrix® (0.49 mmol/g and 0.18 mmol/g loading) resin was purchased from PCAS Biomatrix. Water was deionized using a Milli-Q Reference water purification system (Millipore). Nylon 0.22 μm syringe filters were TISCH brand SPEC17984. 5 mL and 10 mL peptide synthesis reaction vessels were purchased from Torviq (catalog numbers SF-0500, and SF-1000 respectively). Biotin-PEG₁₂-COOH (98%) was purchased from BroadPharm. Syringe tip caps were purchased from VWR (catalog number 97001-202).

### Materials for protein folding

Guanidine hydrochloride (Gdn HCl, molecular biology grade, ≥99%), 4-(2-hydroxyethyl)1-piperazineethanesulfonic acid (HEPES, ≥99.5%), tris(hydroxymethyl)aminomethane (TRIS, ≥99.9%), tris(hydroxymethyl)aminomethane hydrochloride (TRIS HCl, ≥99.0%), monopotassium phosphate (KPi, ≥99.0%), dipotassium phosphate (≥99.0%), sodium chloride (NaCl, BioXtra, ≥99.5%), glycerol (≥99.5%), sodium hydroxide (≥98%), and hydrochloric acid (36.5–38%) were purchased from Sigma Aldrich. A 0.5 M solution of tris(2-carboxyethyl)phosphine hydrochloride (TCEP, Bond-Breaker™, catalogue number 77720), was purchased from Thermo Fisher Scientific. 1,4-Dithio-DL-threitol (DTT, ≥99%) was purchased from Chem-Impex.

### Automated flow peptide synthesis

All peptides were synthesized on one of three automated fast-flow systems built in the Pentelute laboratory. In all cases, synthesis conditions were identical to the optimized protocol as previously described[31]. Every protein target was synthesized once to yield sufficient amounts for analytical and biophysical characterization, with the exception of L-CHIP, L-IRAK2, L-ERG, L-BCL11a, L-Myc, L-Max, L-Max-nb, L-Max-Max-nb, and L-Myc-Max-nb, which were synthesized twice. Conditions are provided in the Supplementary Methods section.

### Manual peptide synthesis

Unless otherwise specified, all manual coupling reactions were carried out in 10 mL disposable peptide synthesis vessels from Torviq attached to vacuum manifolds. To the reaction vessel was added an appropriate amount of resin and DMF was added to completely fill the vessel. The mixture was stirred with a glass rod to remove clumps and fully suspend the resin in DMF. The finely divided resin was incubated in DMF for 30 min to fully swell. To a 500 mL volumetric flask (VWR, part number 10124-384) was weighed 72.24 g (0.19 mol) of HATU and DMF was added to prepare a 0.38 M stock solution. The HATU stock solution was transferred to a 500 mL Corning Pyrex glass media bottle (part number 1395-500) and stored for not more than 1 week. To a 20 mL glass scintillation vial was weighed 5 equivalents (relative to resin loading) of Fmoc-protected amino acid. To the glass scintillation vial was added an appropriate volume of HATU stock to produce a 0.4 M solution of Fmoc-protected amino acid. The amino acid was fully solubilized with sonication and stored for not more than 24 h. In cases where the amino acid was not fully solubilized, DMF was added to adjust amino acid concentration to 0.2 M. After the incubation time DMF was drained from the reaction vessel. To the glass scintillation vial with the solubilized amino acid and HATU was added 15 equivalents of DIEA (relative to resin loading), the screw cap affixed, and the solution mixed rapidly. After a 30 s incubation, the solution containing the now

activated amino acid was added to the reaction vessel. The resin was stirred with the same glass rod to produce a homogeneous mixture, and the apparatus was allowed to sit for 15 min. The reaction mixture was drained from the reaction vessel, and the resin rinsed with 10 mL DMF three times. After washing the resin, the Fmoc protecting group was removed from the terminal amine. Separately, a stock solution of 20% piperidine (v/v) in DMF was prepared in a 500 mL Corning Pyrex glass media bottle. To the reaction vessel was added enough of the deprotection mixture to fully cover the resin, and let sit for 15 min. The deprotection mixture was drained from the reaction vessel and the resin rinsed with 10 mL DMF three times.

## Cleavage protocols
Following synthesis, peptidyl resins were washed with dichloromethane (5 × 5 mL) and dried under an $N_2$ stream. The peptidyl resin was transferred to a 10 mL reaction vessel and stored at 4 °C until cleavage. Special care was taken with peptidyl resins for sequences containing methionine to avoid prolonged storage. Cleavage was performed by one of two protocols (see Supplementary Methods). After cleavage, the solid pellet was dried gently with $N_2$ flow until no visible liquid remained. To the Falcon tube containing the dried peptide precipitate was added a volume equal to 3 times that of the precipitate of a 50:50 mixture of water/acetonitrile containing 0.1% TFA. To aid solubilization of the precipitate, the assembly was re-capped and vortexed. In most cases, the precipitate was fully solubilized. The mixture was left to incubate for 30 min at room temperature. The mixture was then flash-frozen with liquid nitrogen and lyophilized for at least 16 h to provide the freeze-dried crude polypeptide as a white solid.

## Analytical reverse phase high pressure liquid chromatography
Analysis of synthetic peptides was primarily carried out by evaluation of the chromatogram produced from the absorbance at 214 nm after separation on a reverse phase column. In each case, an appropriate amount of material was injected to produce peaks on the chromatogram that were within the dynamic range of the UV detector, typically between 100 and 2000 mAU (3–5 μg of the major product) for the major peak. One of three methods was used for each synthetic protein chain (see Supplementary Methods and Supplementary Results).

## Reverse phase high-performance liquid chromatography–mass spectrometry (LC–MS)
Solutions of the synthetic peptides were evaluated by LC–MS using one of four methods (see Supplementary Methods and Supplementary Results). Data were processed using Agilent MassHunter Workstation Qualitative Analysis Version B.10.00 or Agilent MassHunter BioConfirm B.10.00. Deconvoluted masses of proteins were obtained using a maximum entropy algorithm. Unless otherwise depicted, the following parameters were used for deconvolution: A range 3000 Da less than the starting mass rounded down to the nearest 10,000 Da was used as a lower limit (not lower than 5,000 Da). For the higher limit, a mass to the nearest 10,000 Da rounding up was used. A limited $m/z$ range from 600 to 3000 was used with a baseline subtraction factor of 3 and a mass step of 1. Unless otherwise mentioned, the mass spectrometry data were extracted from the integrated total ion count of the entire major product peak for all targets.

## Preparative reverse phase purification
Crude lyophilized peptide powder was weighed and added to a 50 mL Falcon tube. A denaturing buffer consisting of 6 M Gdn HCl, 50 mM TRIS HCl, 50 mM DTT was freshly prepared, and the pH adjusted with 6 M NaOH and 6 M HCl to a value of 7.5 as measured with a pH electrode. An aliquot of the denaturing buffer was added to the lyophilized peptide to prepare a solution at 10 mg of peptide per mL of buffer. A fresh screw-cap was added to the Falcon tube, and the mixture was

sequentially vortexed and sonicated until no visible precipitate remained. Depending on the solubility of the peptide sequence, additional denaturing buffer was required. The pH was readjusted to 7.5 with 6 M NaOH and 6 M HCl as measured by pH paper, and the mixture was incubated for 30 min at room temperature. After incubation, the Falcon tube screw cap was removed, and the mixture was transferred to a 20 mL syringe affixed with a 0.22 μM nylon syringe filter. The mixture was forced through the syringe filter into a fresh Falcon tube. The Falcon tube originally containing the unfiltered peptide solution was rinsed with an additional 2 mL of the denaturing buffer that was then filtered using the same filtration apparatus into the Falcon tube containing the filtered peptide solution. Any bubbles generated were removed by centrifugation of the falcon tube containing the dissolved peptide at 3220 × g for 5 min. The clarified peptide mixture was drawn-up with a fresh 20 mL syringe and added to the purification column via multiple injections with a manual injection valve affixed with a 10 mL loading loop. Purification was performed in each case by one of three methods (see Supplementary Methods and Supplementary Results).

## Phage display
*Phage Library Construction and Screening.* Stapled phage-displayed peptide libraries were constructed as previously described[46]. Briefly, Peptide Display Cloning System kit from New England Biolabs is used to construct M13KE-containing stapled phage libraries (New England Biolabs, Ipswich, MA). Library oligonucleotides are chemically synthesized using a mix of trimer phosphoramidites (Glen Research, Sterling, VA) without codons encoding cysteine, lysine, proline, and glycine, annealed, extended, and ligated into a digested M13KE vector. All DNA products are purified using Monarch PCR and DNA cleanup kit (New England Biolabs, Ipswich, MA). The resulted library-containing phage vector is transformed into *E. coli* strain ER2738 (Lucigen, Middleton, WI) by electroporation and amplified by adding the post-rescue electroporated cells to a 500 mL *E. coli* culture at early-lag phase (OD600 = 0.01). Phage propagation, purification, and stapling are conducted as described[46]. The quality of the final phage libraries is assessed by Sanger and Next Generation Sequencing. To conduct phage library screening, we followed a previously described procedure[46]. Briefly, peptide-displayed phage libraries are incubated with streptavidin magnetic beads for 1 h at room temperature in a blocking buffer containing 5% w/v nonfat milk. For each screening condition, biotinylated protein is captured with streptavidin-coated magnetic beads at room temperature for 15 min, the supernatant is removed using a plate magnet and the beads are resuspended in 50 μL of the blocking buffer. 150 μL of the depleted phage library is added to each well for 200 μL final volume, plates are sealed, and the screening reactions are incubated at room temperature for 45 min, with rotation to maintain beads in solution. Following binding, beads are washed 5 times with ice-cold washing buffer (1X TBS, 1 mM $MgCl_2$, 1% (w/v) BSA, 0.1% Tween-20, 0.02% (w/v) sodium azide, 2% (w/v) glycerol), beads containing protein-bound phage are collected and directly processed for NGS.

*Phage Next Generation Sequencing, Hit ID and Clustering.* Next Generation Sequencing was performed according to a previously described procedure[46]. Briefly, phage particles are denatured from magnetic beads at 95 °C for 15 min with an added spike-in sequence (not a library member that is used to enable cross-well normalization of sequence reads), followed by a two-step low-cycled PCR to introduce Illumina adaptors and 10 bp TruSeq DNA UD Indexes (Illumina, San Diego, CA) according to an Illumina's 16S Metagenomic Sequencing Library Preparation protocol. The NGS library is sequenced by an Illumina NovaSeq platform using a 2 × 150 bp high-output kit (Illumina, San Diego, CA). Hit ID and clustering was performed as previously described[46]. Briefly, NGS reads are trimmed for quality and filtered for sequences that matched the design of the phage library. Counts for

each unique sequence are tallied, and then normalized by the counts of the spike-in sequence added to each sample. A metric called Hit Strength is computed for each sequence as the fold change between the normalized counts in the highest target concentration sample and the normalized counts in the blank samples (averaged across experimental replicates). When 0 counts are observed for a sequence in blank samples, a count of 0.5 is used to prevent dividing by zero.

### Determination of protein concentration

The concentration of proteins in aqueous solutions was determined using the solution's absorbance at 280 nm ($A_{280}$), or the Quick Start™ Bradford Protein Assay (see Supplementary Methods).

### Protein folding via size exclusion chromatography

Semi-preparative size exclusion chromatography for the folding of synthetic proteins was carried out on an Agilent InfinityLab LC Series 1260 Infinity II Bio-Inert LC System with a Superdex 75 Increase 10/300 GL size exclusion chromatography (SEC) column. In all cases, the flow rate used was 0.8 mL/min. The sample application was accomplished with the use of an Agilent 1200 series manual injection valve fitted with either a 0.5 mL or 0.25 mL loading loop placed before the entrance to the SEC column. An appropriate running buffer was freshly prepared and filtered through a 0.22 μm PES bottle top filter (Corning, catalog number 431118) and the SEC column equilibrated in the running buffer for at least 2 column volumes before sample application. Unless otherwise specified, the running buffer was 50 mM HEPES, 150 mM NaCl, 0.5 mM DTT, 5% glycerol (v/v) adjusted to pH 7.5 with 6 M NaOH or 6 M HCl as required.

An appropriate amount of lyophilized, purified protein (typically 0.1–1.0 mg) was weighed into a low-protein-binding 2 mL Eppendorf tube. An appropriate denaturing buffer was freshly prepared. Unless otherwise specified, the denaturing buffer consisted of 6 M Gdn HCl, 50 mM TRIS HCl, and 50 mM DTT adjusted to pH 7.5 with 6 M NaOH or 6 M HCl as required. A sufficient amount of denaturing buffer was added to the protein to achieve a final concentration of 10 mg/mL. Buffer was added as required to achieve a completely clear solution with no precipitate, but the total volume of the sample was kept below half the volume of the loading loop. The sample was clarified by centrifugation at 21,000 × g for 15 min. The sample was applied to the SEC column and eluted over 2 column volumes with manual fractionation into 2 mL low-protein-binding Eppendorf tubes. Fractions corresponding to the folded protein were collected, and checked by LC-MS. The resulting mixture was separated into 1 nmol aliquots, flash frozen with liquid nitrogen, and stored at −80 °C until further use.

### Protein folding via dilution

An appropriate amount of lyophilized, purified protein (typically 0.1–1.0 mg) was weighed into a low-protein-binding 5 mL Eppendorf tube. Appropriate volumes of a denaturing buffer and a dilution buffer were freshly prepared. Unless otherwise specified, the denaturing buffer consisted of 6 M Gdn HCl, 50 mM TRIS HCl, and 50 mM DTT adjusted to pH 7.5 with 6 M NaOH or 6 M HCl as required, and the dilution buffer consisted of 50 mM HEPES, 150 mM NaCl, 0.5 mM DTT, 5% glycerol (v/v) adjusted to pH 7.5 with 6 M NaOH or 6 M HCl as required. Denaturing buffer was added to the protein to achieve a final concentration of 10 mg/mL. Protein concentration was measured and the stock concentration adjusted with additional denaturing buffer to 150 μM. Dilution buffer was added to the protein in denaturing buffer to afford a final protein concentration of 25 μM. The resulting solution was kept at room temperature for 1 h before further use.

### Software

Data collection was performed with the assistance of Agilent MassHunter (Version B.06.01), Agilent ChemStation (Version C.01.10[287]), AVIV CD control software, Tecan SparkControl (Version 3.1), PheraStar Software (V 5.70 R4), Biacore S200 Control Software (V 1.1), Biacore 8K Control Software (V 3.0.12.15655), and CLARIOstar Software (V.5.40 R3). Data analysis was performed with the assistance of Agilent MassHunter Workstation Qualitative Analysis (Version B.10.00), Agilent MassHunter BioConfirm Software (Version B.10.00), GraphPad Prism (Version 9.4.0), Agilent ChemStation (Version C.01.10[287]), Adobe Illustrator (Version 27.4.1), Microsoft Excel (2019, Version 1808), ChemDraw (Version 19.1.0.8), PheraStar MARS (V 3.42 R5), BIAevaluation (V 4.1), and CLARIOstar (V.5.40 R3).

### Reporting summary

Further information on research design is available in the Nature Portfolio Reporting Summary linked to this article.

## Data availability

All data are available in the main text or the supplementary materials (Supplementary Information and Data Source files). Crystallographic datasets have been deposited in the Protein Data Bank (PDB IDs: 8F14, 8F15, 8F16, 8F17, 8F0Z, 8F12, 8F13, 8F10). The raw sequencing data from mirror-image phage display selections against MDM2 and CHIP was deposited with the NCBI Sequence Read Archive (SRA) and can be retrieved using BioProject accession code PRJNA997450 (ID: 997450) and BioSample accession codes SAMN36679786 and SAMN36679787. Source data are provided with this paper.

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

## Acknowledgements

Financial support for this work was provided by FOG Pharmaceuticals (to B.L.P.). We thank Mirella Bucci and Jacob Rodriguez for their valuable comments during manuscript preparation. Please refer to Supplementary Information Section 3 for acknowledgments related to X-ray crystallography data collection.

## Author contributions

A.J.C., G.L.V., J.H.M., and B.L.P. conceptualized the research. A.J.C., S.G., R.M.R., L.L.S., S.H., and Y.C.L. carried out protein synthesis and A.J.C. carried out protein folding. T.L.T. and A.J.C. carried out protein biochemical validation. O.S.T., K.L., and J.M.S. carried out phage screening and binder validation. K.L. carried out protein crystallization and structure refinement. A.J.C., S.G., R.M.R., and K.L. generated figures. A.J.C., A.L., J.H.M., and B.L.P. wrote the manuscript with input from all authors.

## Competing interests

The authors declare the following competing interest(s): B.L.P. is a co-founder and/or member of the scientific advisory board of several companies focusing on the development of protein and peptide therapeutics. T.L.T., K.L., O.S.T., G.L.V., and J.H.M. are currently employed by FOG Pharmaceuticals, and G.L.V. serves on the board of directors of FOG Pharmaceuticals. FOG Pharmaceuticals has filed provisional patent applications related to the compounds described in this work. J.M.S. was an employee of FOG Pharmaceuticals and is currently employed by Relay Therapeutics, Inc.
