## [Peer Review File · Nature Communications]

REVIEWER COMMENTS

Reviewer #1 (Remarks to the Author):

D-peptides and proteins have the intrinsic advantages for therapeutic applications because of the high in vivo stability, low immunogenicity properties. However, these molecules haven't really lived up to the promise in terms of practical applications. As the authors in this manuscript pointed out, the bottleneck for accessing these D-peptides/proteins therapeutics is the efficient preparation of the D-protein targets. Thus, in this current manuscript, the authors try to streamline the discovery of D-peptides by establishing the efficient synthesis of D-proteins and selecting D-peptides using mirror image phage display.

Highlights in the current manuscript:

1. It is quite impressive that the authors were able to show the successful synthesis of 24 proteins (70-132 amino acids, 12 L proteins and 12 corresponding D-proteins) in this single manuscript. This is not often seen in literatures.
2. The fact that the authors were able to prepare all these proteins using similar flow synthesis method is also impressive, although from the data shown in the main text and SI, we can see some syntheses are better than others. I am not sure how many attempts the authors made towards each protein synthesis, I would guess there still needs to be some optimizations done for different proteins. It would be helpful if the authors can discuss how many attempts they made for every protein synthesis, what are the optimizations they did if initial synthesis was not satisfactory. This detailed information would be valuable for people working in this area.
3. The authors did phage selections against MDM2 and CHIP, they were able to get binders with reasonable binding affinity. They also determined some binder-target complex crystal structures to further demonstrate the success of the selection.

Limits in this manuscript:

1. Flow synthesis method has been well reported by the same group, they have also shown flow synthesis can access small proteins up to ~100-200 amino acids prior current manuscript. In the 2020 Science paper, the group reported 9 protein synthesis using flow method. Thus, in terms of synthetic novelty, current manuscript doesn't present new things here.
2. The general mirror image phage display pipeline has long been established; many different groups have also adopted this pipeline to access D-peptides/proteins in the past. In terms of the general pipeline, the current manuscript doesn't bring in too many new things. I do acknowledge that better access to D-protein targets could facilitate D-peptides/proteins discovery.
3. Many protein target size will go beyond 200 amino acids or 300 amino acids, I would think flow synthesis will still need to be combined with ligation methods to access these bigger target proteins, especially if people want to access these targets with high purity and homogeneity, high quality target preparation is key to the success of final D-peptide discovery, the authors probably should discuss this aspect.
4. For generating D-peptides, there is a phage selection step after the D-target preparation. This current manuscript showed two examples of successful selections, but these are routine examples, couldn't contribute much to the novelty or significance of this manuscript.

Other comments:

1. Line 76-94, the authors tried to convey the message that modern chemical protein synthesis has to do individual optimization for each target while flow synthesis doesn't. I think the authors really need to make the same set of proteins prepared by others to make this statement. Many reported "difficult proteins" were not repeated by the authors using flow synthesis; thus, this statement may not stand. Like I mentioned previously, it would be helpful if the authors can discuss how many attempts they made for every protein synthesis, what are the optimizations they did if one synthesis is not satisfactory. This detailed information would be valuable for people working in this area.
2. For Fig. 2, If the mass data corresponds to the entire HPLC peak, then the authors can shade the

HPLC peak to show it. If it's not the entire HPLC peak, then they need to show the mass data which corresponds to the entire HPLC peak.

3. The authors need to show the crude protein synthesis LC-MS analysis in the SI for all proteins, not just the purified product. They also need to replace all HPLC and LC-MS data in the SI with high resolution figures, it's difficult to read the current figures in the SI. I think they need to show high resolution mass data with resolution labeled (± 1 Da?), the proteins are not very big (less than 20 KDa), so the deconvoluted product mass should have a sharp peak if it's pure enough.

4. In Fig. 4C, for SPR experiments, the authors should show the original binding curves at different peptide concentration (association and dissociation).

5. The claims in line 409-410 "We demonstrate for the first time a robust and scalable platform for D-peptide discovery using mirror-image phage display facilitated by automated flow synthesis technology." And Line 434-435. "Collectively, the results reported here represent a significant fraction of the reported successful mirror-image phage display screens." might need to be modified. The authors did show 12 D-protein syntheses here, however, they only showed selection for binders against two targets and similar work for MDM2 has been done in the past by other groups as the author pointed out.

Overall, I think the manuscript is a good demonstration that D-peptides discovery could be greatly facilitated if D-targets could be better accessed. Although the novelty of the current manuscript is moderate (all related chemistry or technology has been established in the past), the impact of this work could be significant. Thus, the manuscript could be reconsidered after revision.

Reviewer #2 (Remarks to the Author):

I was asked by the editor to specifically only comment on the reported co-crystal structures reported by Callahan et al. in the manuscript "Mirror-Image Ligand Discovery Enabled by Single-Shot Fast-Flow Synthesis of D-Proteins".

The crystal structures are overall of good quality and the electron density maps support the modelled co-crystal structures with few exceptions as outlined below. The authors should address the following issues before acceptance of the manuscript.

I) In the co-crystal structures of Mdm2 with the D-H103 peptide (PDB IDs 8F12 and 8F13) all D-methionine residues appear to be incorrectly modelled as D-norleucine (PDB chemical ID DNE). These need to be changed to the correct PDB chemical ID (MED) and the models re-refined against the data.
II) The electron density maps of the co-crystal structure of the CHIP TPR domain in complex with D-H203 (PDB 8F16) show clear and strong additional density for the N-term of peptide, suggesting that the peptide is present in different conformations. This is not adequately modelled in the current crystal structure and needs to be added and re-refined. Are the two different conformations relevant for the function of the peptide?

III) In the co-crystal structure of CHIP with the D-H203 peptide (PDB 8F17), the N-terminal acetyl group and dPro1 have incorrect chain IDs or residue numbers, leading to inconsistencies in structure visualisation. Specifically, the N-terminal acetyl group (ACE) of chain D has chain ID C, and the dPro has residue number 101 instead of 2.

IV) What is the difference between the structures MDM2-D-H103 and MDM2-D-H103-Alt? The latter is listed in the supplement but not discussed in the manuscript.

V) Why did the authors not discuss the structure of CHIP E3 ligase with D-H204 peptide?

VI) Protein/peptide complex concentrations (in mg/ml or μ M) used for crystallisation should be indicated in Section 11 of the supplement to allow better reproducibility.

Reviewer #3 (Remarks to the Author):

This manuscript from Pentelute and co-workers describes the use of fast flow peptide synthesis developed by the group (and published in a 2020 Science paper) to generate 12 small D-proteins in a single synthetic run. These mirror image proteins were subsequently folded and the correct fold validated by binding experiments to known ligands (generated in their mirror image forms). From here the authors took two of the D-proteins – the E3 ligases MDM2 and CHIP - and performed mirror image phage display to arrive at several D-macrocytic peptides with single digit micromolar affinity for the native L-targets. Overall this is an impressive body of work and would be of significant interest to the readership of Nature Communications. I would recommend publication after the minor revisions/corrections outlined below:

1) In some respects, the mirror image display on only 2 proteins after the synthesis of 12 D-proteins weakened the narrative of the paper but this would clearly be a significant amount of extra work to do all of them. It would be useful for the authors to articulate why these particular D-proteins of the 12 were chosen. The authors describe MDM2 as a benchmarking example but what about CHIP?

2) Line 35, page 2: "in a single pass" could be modified to "in a single run" or similar to help the non-specialist reader

3) "With unhindered access to mirror-image protein targets". The word unhindered should be deleted as this is an overstatement given the number and size of proteins prepared in the study.

4) line 76/77, page 3 – the authors could consider citing a recent review on mirror image protein synthesis here (Nature Reviews Chemistry 2023, 7, 383–404)

5) line 104, page 6: "...peptides in a single screening pass." It is not clear what the authors mean by "single screening pass". Recommend rewording to make this clear to the reader.

6) Figure 2 displays the core data of the paper for 12 proteins in L and D-form. The HPLC chromatograms are offset in the Figure but it would be much better if these could be arranged in a way where it is clear to see that the two proteins have identical retention times. The deconvoluted mass specs also offset and again it would be better to show that the masses line up. I would recommend removing the e6 values on the x axis of the deconvoluted spectra and instead use kDa as the x axis label that would allow whole numbers to be quoted which will be simpler for the reader. If possible it would also be useful if the ESI mass specs and deconvoluted mass specs could be made slightly larger.

7) line 246, page 10: typo "assess"

8) line 289, page 11: "carried it into our next-generation MIPD platform". As far as I could see the mirror image display that was performed was standard so "next-generation" should be deleted.

9) Line 323, page 13: How do the affinities of the MDM2 ligands generated in this MIPD campaign compare to those generated by MIPD previously? It would be great if the authors could unpack any advantages in the macrocytic scaffold vs linear peptide ligands from MIPD.

10) line 400, page 16: Recommend replacing "prevented" with "complicated" given that there are plenty of linear L-peptide drugs in the clinic.

11) The authors have included a very comprehensive Supplementary Information file to accompany the manuscript. Some points of clarification on data are provided below.

i) Is the mass spectral data extracted from the total ion count from the whole LCMS experiment? This information should be added somewhere in the figure/legend

ii) PgS22 – what is the additional set of ions in the IRAK2 mass spectrum?

iii) PgsS27, S28, S33 – what are the impurities in the D-NEMO, L-FKBP12, D-YAP1 proteins that are evident from the mass spectra?

Bradley L. Pentelute
Professor
Phone: (617) 324-0180
Email: blp@mit.edu

Massachusetts Institute of Technology
Department of Chemistry, Room 18-596
77 Massachusetts Avenue
Cambridge, MA 02139-4307

November 8, 2023

Dear Reviewers,

We thank you for your review of our manuscript entitled “*Mirror-Image Ligand Discovery Enabled by Single-Shot Fast-Flow Synthesis of D-Proteins*” which we submitted to *Nature Communications* (manuscript NCOMMS-23-29181-T). We appreciate your efforts and are grateful for your insightful comments. In our point-by-point responses below, the reviewers’ remarks are reproduced in black color, whereas **our responses are given in green color**.

Reviewer 1 (Remarks to the Author)

D-peptides and proteins have the intrinsic advantages for therapeutic applications because of the high in vivo stability, low immunogenicity properties. However, these molecules haven’t really lived up to the promise in terms of practical applications. As the authors in this manuscript pointed out, the bottleneck for accessing these D-peptides/proteins therapeutics is the efficient preparation of the D-protein targets. Thus, in this current manuscript, the authors try to streamline the discovery of D-peptides by establishing the efficient synthesis of D-proteins and selecting D-peptides using mirror image phage display.

Highlights in the current manuscript:

1. It is quite impressive that the authors were able to show the successful synthesis of 24 proteins (70-132 amino acids, 12 L proteins and 12 corresponding D-proteins) in this single manuscript. This is not often seen in literatures.

We thank the reviewer for their appreciation of our work.

2. The fact that the authors were able to prepare all these proteins using similar flow synthesis method is also impressive, although from the data shown in the main text and SI, we can see some syntheses are better than others. I am not sure how many attempts the authors made towards each protein synthesis, I would guess there still needs to be some optimizations done for different proteins. It would be helpful if the authors can discuss how many attempts they made for every protein synthesis, what are the optimizations they did if initial synthesis was not satisfactory. This detailed information would be valuable for people working in this area.

Each target protein was initially synthesized once for the original manuscript submission according to the protocol established in our previous publication (Science **368, 980 (2020)), yielding sufficient amounts post-purification for all subsequent analytical and biophysical characterization studies. No changes or further optimizations were made to the protocol described above in any of the cases.**

In order to address the reviewer’s comments regarding suitability of the general protocol for reproducible outcomes, and also to gather additional analytical data for the crude material comparisons requested below (see Other Comment 3, page 3), we repeated the syntheses of half of the proteins in L-forms (L-CHIP, L-IRAK2, L-ERG, L-BCL11a, L-Myc, L-Max, and L-Max-nb), as well as repeated the assembly of the L-Max-Max and L-Myc-Max covalent dimers. In all cases, the repeated syntheses proceeded with a high degree of reproducibility as monitored by the UV-

Bradley L. Pentelute
Professor
Phone: (617) 324-0180
Email: blp@mit.edu

Massachusetts Institute of Technology
Department of Chemistry, Room 18-596
77 Massachusetts Avenue
Cambridge, MA 02139-4307

Vis Fmoc deprotection traces, and the analytical HPLC and LC-MS characterization data pre- and post-purification closely resembled our initially reported spectra. Per reviewer's request in a separate comment below, we also added the analytical HPLC traces of all crude materials in the Supplementary Information file, sections 5.1 and 5.2.

3. The authors did phage selections against MDM2 and CHIP, they were able to get binders with reasonable binding affinity. They also determined some binder-target complex crystal structures to further demonstrate the success of the selection.

We thank the reviewer for their supportive comments and address their concerns below.

Limits in this manuscript:

1. Flow synthesis method has been well reported by the same group, they have also shown flow synthesis can access small proteins up to ~100-200 amino acids prior current manuscript. In the 2020 Science paper, the group reported 9 protein synthesis using flow method. Thus, in terms of synthetic novelty, current manuscript doesn't present new things here.

Our earlier 2020 Science paper focused on accessing L-proteins and arrived at the optimized synthesis protocol which was employed largely unchanged in the current study. We recognize that our current manuscript does not present any technological advances or chemistry optimization related to the polypeptide chain synthesis protocol. However, this work reports the extension of the flow synthesis protocol to D-amino acids as a general method to rapidly access unnatural D-proteins through a single-shot approach, demonstrating the utility of our technology to deliver mirror-image variants of native proteins. Furthermore, we provide compelling evidence of the utility of these D-proteins in mirror-image phage display and demonstrate the identification of proteolytically stable D-peptide binders for two pivotal targets.

2. The general mirror image phage display pipeline has long been established; many different groups have also adopted this pipeline to access D-peptides/proteins in the past. In terms of the general pipeline, the current manuscript doesn't bring in too many new things. I do acknowledge that better access to D-protein targets could facilitate D-peptides/proteins discovery.

We thank the reviewer for highlighting the outcome of our work. As the reviewer mentioned, the efficient synthesis of D-protein targets is key to development of efficient D-peptide binders by MIPD. In this work, we present an efficient approach for rapid access to mirror-image proteins. As outlined in this manuscript, we report the production of 12 different D-proteins, encompassing approximately one third of all D-proteins reported to date, including 8 new targets.

3. Many protein target size will go beyond 200 amino acids or 300 amino acids, I would think flow synthesis will still need to be combined with ligation methods to access these bigger target proteins, especially if people want to access these targets with high purity and homogeneity, high quality target preparation is key to the success of final D-peptide discovery, the authors probably should discuss this aspect.

Bradley L. Pentelute

Professor

Phone: (617) 324-0180

Email: blp@mit.edu

Massachusetts Institute of Technology

Department of Chemistry, Room 18-596

77 Massachusetts Avenue

Cambridge, MA 02139-4307

We completely agree with the reviewer's perspective, emphasizing the need of combining the flow synthesis with ligation methods to access large and difficult protein sequences. Our ongoing efforts are focused on combining these two powerful approaches to arrive at synthetic functional proteins exceeding 200 amino acids, and we anticipate reporting the outcomes in the near future.

4. For generating D-peptides, there is a phage selection step after the D-target preparation. This current manuscript showed two examples of successful selections, but these are routine examples, couldn't contribute much to the novelty or significance of this manuscript.

The primary goal of the current manuscript is to emphasize the single-shot flow synthesis approach to access functional D-protein targets rapidly. Among these, we chose 2 important ubiquitin ligases to evaluate with a recently reported mirror-image phage display screening platform to generate high-affinity conformationally constrained α -helical peptides. Notably, even though the screening platform largely follows established protocols, this manuscript presents the first report on the development of D-peptide binders targeting the CHIP protein.

Other comments:

1. Line 76-94, the authors tried to convey the message that modern chemical protein synthesis has to do individual optimization for each target while flow synthesis doesn't. I think the authors really need to make the same set of proteins prepared by others to make this statement. Many reported "difficult proteins" were not repeated by the authors using flow synthesis; thus, this statement may not stand. Like I mentioned previously, it would be helpful if the authors can discuss how many attempts they made for every protein synthesis, what are the optimizations they did if one synthesis is not satisfactory. This detailed information would be valuable for people working in this area.

We appreciate the reviewer's observation. It was not our goal to make a head-to-head comparison of all reported "difficult" protein syntheses with our flow instrumentation. In the mentioned lines of text, we simply wanted to emphasize recognized challenges in the chemical protein synthesis field. To date, the majority of the synthetic protein targets were synthesized by a combination of solid phase peptide synthesis and ligation methods. The ligation part, in particular, requires the identification of optimal conditions to find suitable ligation junctions and increase conversion to the desired ligation product. In the current manuscript, we have effectively showcased the single-shot synthesis of Barnase, bypassing the need for ligation strategies (*ChemBioChem* **2014**, *15*(5), 721-733 and *Chem. Sci.* **2015**, *6*(5), 2997-3002).

We acknowledge individual optimization may be needed for the flow synthesis of certain sequences, which are identified in our laboratory on a case-by-case basis according to on-line UV-vis monitoring of the stepwise synthesis cycle and the quality of the crude material obtained. However, this was not the case here as each target protein was synthesized successfully once on the initial attempts according to the protocol established in our previous publication (*Science* **2020**, *368*, 980). As mentioned above in our response to Highlight 2 (page 1), no changes or further optimizations were made to the protocol described.

Bradley L. Pentelute

Professor

Phone: (617) 324-0180

Email: blp@mit.edu

Massachusetts Institute of Technology

Department of Chemistry, Room 18-596

77 Massachusetts Avenue

Cambridge, MA 02139-4307

2. For Fig. 2, If the mass data corresponds to the entire HPLC peak, then the authors can shade the HPLC peak to show it. If it's not the entire HPLC peak, then they need to show the mass data which corresponds to the entire HPLC peak.

The mass data presented in the insets correspond to the major HPLC peak for all target proteins reported in this manuscript. The LC-MS analyses were performed separately by carrying out injections from the same sample solution used to generate the HPLC chromatograms. Asterisks were added to indicate the mass spectra shown correspond to the designated HPLC peaks. We have also included this information in the footnotes of Figure 2 in the main manuscript.

3. The authors need to show the crude protein synthesis LC-MS analysis in the SI for all proteins, not just the purified product. They also need to replace all HPLC and LC-MS data in the SI with high resolution figures, it's difficult to read the current figures in the SI. I think they need to show high resolution mass data with resolution labeled (+/- 1 Da?), the proteins are not very big (less than 20 KDa), so the deconvoluted product mass should have a sharp peak if it's pure enough.

We have included the crude protein synthesis data for all target proteins in Section 5.1 and 5.2 of the supplementary information file. As the reviewer suggested, we have replaced all HPLC and LC-MS data with higher resolution figures, and we have included the corresponding mass data for each target along with an expanded spectrum of the deconvoluted mass peak. It's important to note that the broadening observed for some product peaks in the mass spectra is a result of sodium (Na⁺) adducts from the LC-MS and are not due to impurities.

4. In Fig. 4C, for SPR experiments, the authors should show the original binding curves at different peptide concentration (association and dissociation).

The binding curves at different concentrations (association and dissociation) are shown in Figures S6-S8 in the Supplementary Information file.

5. The claims in line 409-410 "We demonstrate for the first time a robust and scalable platform for D-peptide discovery using mirror-image phage display facilitated by automated flow synthesis technology." And Line 434-435. "Collectively, the results reported here represent a significant fraction of the reported successful mirror-image phage display screens." might need to be modified. The authors did show 12 D-protein syntheses here, however, they only showed selection for binders against two targets and similar work for MDM2 has been done in the past by other groups as the author pointed out.

We thank the reviewer for the suggestion. We modified the sentences as: "We demonstrate a robust and scalable platform that uses automated flow synthesis technology to facilitate access to mirror-image proteins for D-peptide discovery using mirror-image phage display" in lines 409-410, and: "Collectively, the results reported here represent a significant fraction of the reported successful D-protein syntheses to date" in lines 434-435.

Overall, I think the manuscript is a good demonstration that D-peptides discovery could be greatly facilitated if D-targets could be better accessed. Although the novelty of the current manuscript is

Bradley L. Pentelute
Professor
Phone: (617) 324-0180
Email: blp@mit.edu

Massachusetts Institute of Technology
Department of Chemistry, Room 18-596
77 Massachusetts Avenue
Cambridge, MA 02139-4307

moderate (all related chemistry or technology has been established in the past), the impact of this work could be significant. Thus, the manuscript could be reconsidered after revision.

Reviewer #2 (Remarks to the Author)

I was asked by the editor to specifically only comment on the reported co-crystal structures reported by Callahan et al. in the manuscript “Mirror-Image Ligand Discovery Enabled by Single-Shot Fast-Flow Synthesis of D-Proteins”.

The crystal structures are overall of good quality and the electron density maps support the modelled co-crystal structures with few exceptions as outlined below. The authors should address the following issues before acceptance of the manuscript.

1) In the co-crystal structures of Mdm2 with the D-H103 peptide (PDB IDs 8F12 and 8F13) all D-methionine residues appear to be incorrectly modelled as D-norleucine (PDB chemical ID DNE). These need to be changed to the correct PDB chemical ID (MED) and the models re-refined against the data.

We thank the reviewer for pointing out this error. Upon reevaluation, we discovered that we had screened the peptide library containing norleucine monomers against the MDM-2 target, and inadvertently provided the incorrect peptide binder sequence in Figure 4. We have updated and corrected the peptide sequence in Figure 4 to accurately reflect the inclusion of norleucine.

2) The electron density maps of the co-crystal structure of the CHIP TPR domain in complex with D-H203 (PDB 8F16) show clear and strong additional density for the N-term of peptide, suggesting that the peptide is present in different conformations. This is not adequately modelled in the current crystal structure and needs to be added and re-refined. Are the two different conformations relevant for the function of the peptide?

We noticed the extra density near the N-term of the peptides, especially around D-His3, D-Glu4, and D-Met5. However, when we attempted to model an alternative conformation, the geometry of the alternative conformation was not acceptable (see below). Thus, we left the density unmodelled without adding alternative conformation or solvent molecules.

Bradley L. Pentelute
Professor
Phone: (617) 324-0180
Email: blp@mit.edu

Massachusetts Institute of Technology
Department of Chemistry, Room 18-596
77 Massachusetts Avenue
Cambridge, MA 02139-4307

3) In the co-crystal structure of CHIP with the D-H203 peptide (PDB 8F17), the N-terminal acetyl group and dPro1 have incorrect chain IDs or residue numbers, leading to inconsistencies in structure visualisation. Specifically, the N-terminal acetyl group (ACE) of chain D has chain ID C, and the dPro has residue number 101 instead of 2.

The PDB record was updated to show the correct structure.

4) What is the difference between the structures MDM2-D-H103 and MDM2-D-H103-Alt? The latter is listed in the supplement but not discussed in the manuscript.

MDM2-D-H103 and MDM2-D-H103-Alt are crystallized under the same space group but under different pH/crystallization conditions. A subtle difference is observed near the C-term (last 2-3 residues) of the D-peptides. The structure with a more helical C-term (MDM2-D-H103) was used for the figures presented in the manuscript.

5) Why did the authors not discuss the structure of CHIP E3 ligase with D-H204 peptide?

The sequences of H203 and H204 are given below:

H203 (1.56 Å)	Ac- HEMCYWADAYCRYS-NH ₂
H204 (2.21 Å)	Ac-PLDLCYWASLHCIVS-NH ₂

H203 and H204 are identified from the same phage cluster and share a similar binding pose. We initially focused on the H203 structure for discussion in the manuscript since it has a higher resolution. Per reviewer's suggestion, we added a brief discussion on the structure of H204 in the manuscript: "We also solved the co-crystal structure of CHIP in complex with D-H204 at a resolution of 2.21 Å. We identified D-H203 and D-H204 from the same phage cluster. As expected, they share a similar amino acid sequence and binding mode."

Bradley L. Pentelute
Professor
Phone: (617) 324-0180
Email: blp@mit.edu

Massachusetts Institute of Technology
Department of Chemistry, Room 18-596
77 Massachusetts Avenue
Cambridge, MA 02139-4307

6) Protein/peptide complex concentrations (in mg/ml or μM) used for crystallisation should be indicated in Section 11 of the supplement to allow better reproducibility.

We have provided these details in Section 11 of the supporting information as the following text: “To obtain the structures of the protein-peptide complexes, briefly, 10 mM peptides stock in 90% DMSO were added to the protein stocks to a final 1:1.25 protein:peptide molar ratio and screened against commercially available crystallization screens”.

Reviewer #3 (Remarks to the Author)

This manuscript from Pentelute and co-workers describes the use of fast flow peptide synthesis developed by the group (and published in a 2020 Science paper) to generate 12 small D-proteins in a single synthetic run. These mirror image proteins were subsequently folded and the correct fold validated by binding experiments to known ligands (generated in their mirror image forms). From here the authors took two of the D-proteins – the E3 ligases MDM2 and CHIP - and performed mirror image phage display to arrive at several D-macrocytic peptides with single digit micromolar affinity for the native L-targets. Overall this is an impressive body of work and would be of significant interest to the readership of Nature Communications. I would recommend publication after the minor revisions/corrections outlined below:

We thank reviewer 3 for their favorable recommendation and address their comments below.

1) In some respects, the mirror image display on only 2 proteins after the synthesis of 12 D-proteins weakened the narrative of the paper but this would clearly be a significant amount of extra work to do all of them. It would be useful for the authors to articulate why these particular D-proteins of the 12 were chosen. The authors describe MDM2 as a benchmarking example but what about CHIP?

The primary objective of the current manuscript is to emphasize the single-shot flow synthesis approach to access D-protein targets rapidly. We successfully demonstrated the single-shot approach for 12 D-protein targets and chose 2 important ubiquitin ligases to carry forward into mirror-image phage display using a recently reported screening platform to generate high-affinity conformationally constrained α -helical peptides. As the reviewer notes, we selected MDM2 as a benchmark with several previously reported macrocytic peptide binders, and chose CHIP as a novel target against which we discovered macrocytic D-peptide binders.

2) Line 35, page 2: “in a single pass” could be modified to “in a single run” or similar to help the non-specialist reader

We have modified the text per the reviewer’s suggestion.

3) “With unhindered access to mirror-image protein targets”. The word unhindered should be deleted as this is an overstatement given the number and size of proteins prepared in the study.

Bradley L. Pentelute
Professor
Phone: (617) 324-0180
Email: blp@mit.edu

Massachusetts Institute of Technology
Department of Chemistry, Room 18-596
77 Massachusetts Avenue
Cambridge, MA 02139-4307

We have modified the text per the reviewer's suggestion.

4) line 76/77, page 3 – the authors could consider citing a recent review on mirror image protein synthesis here (Nature Reviews Chemistry 2023, 7, 383–404)

We thank the reviewer for notifying us, we have included this citation as reference #21 in the revised manuscript.

5) line 104, page 6: “.....peptides in a single screening pass.” It is not clear what the authors mean by “single screening pass”. Recommend rewording to make this clear to the reader.

We have modified the text to: “We utilized a recently reported screening platform based on phage display to generate high-affinity and conformationally constrained α -helical peptide binders’.

6) Figure 2 displays the core data of the paper for 12 proteins in L and D-form. The HPLC chromatograms are offset in the Figure but it would be much better if these could be arranged in a way where it is clear to see that the two proteins have identical retention times. The deconvoluted mass specs also offset and again it would be better to show that the masses line up. I would recommend removing the e6 values on the x axis of the deconvoluted spectra and instead use kDa as the x axis label that would allow whole numbers to be quoted which will be simpler for the reader. If possible it would also be useful if the ESI mass specs and deconvoluted mass specs could be made slightly larger.

We have removed the offset for both the HPLC chromatograms and deconvoluted masses in the revised Figure 2. These data are now depicted as overlapping spectra. Per the reviewer's suggestion, we have replaced the e6 values with kDa on the x axis, and we have increased the size of the ESI and deconvolution masses for better visibility. The data presented in the supporting information file for all proteins provides further details for the analytical HPLC data and deconvoluted mass spectra.

7) line 246, page 10: typo “assess”

We have corrected the typo.

8) line 289, page 11: “carried it into our next-generation MIPD platform”. As far as I could see the mirror image display that was performed was standard so “next-generation” should be deleted.

We have modified per the reviewer's suggestion.

9) Line 323, page 13: How do the affinities of the MDM2 ligands generated in this MIPD campaign compare to those generated by MIPD previously? It would be great if the authors could unpack any advantages in the macrocyclic scaffold vs linear peptide ligands from MIPD.

The previously reported MIPD for MDM2 (*Proc. Natl. Acad. Sci. U. S. A.* **2010**, 107(32), 14321-14326, <https://doi.org/10.1073/pnas.1008930107>) generated linear helical peptides with a higher

Bradley L. Pentelute
Professor
Phone: (617) 324-0180
Email: blp@mit.edu

Massachusetts Institute of Technology
Department of Chemistry, Room 18-596
77 Massachusetts Avenue
Cambridge, MA 02139-4307

affinity, e.g., DPML- α /y with binding affinity of 219/53 nM. The recently reported macrocyclic MDM2-binding peptides from DNA-encoded libraries, UNP-6457, inhibits MDM2-p53 interaction with an IC₅₀ of 8.9 nM (ACS Med. Chem. Lett. **2023**, 14(6), 820–826, <https://pubs.acs.org/doi/10.1021/acsmchemlett.3c00117>). In general, the macrocyclic scaffolds exhibit structural rigidity, enhanced selectivity and resistance to proteolysis, which contribute to improved stability and binding affinity, as well as longer half-lives.

10) line 400, page 16: Recommend replacing “prevented” with “complicated” given that there are plenty of linear L-peptide drugs in the clinic.

We have modified per the reviewer’s suggestion.

11) The authors have included a very comprehensive Supplementary Information file to accompany the manuscript. Some points of clarification on data are provided below.

i) Is the mass spectral data extracted from the total ion count from the whole LCMS experiment? This information should be added somewhere in the figure/legend

We have added a sentence in the revised section 2.5 in Supplementary Information file: “Unless otherwise mentioned, the mass spectrometry data were extracted from the integrated total ion count of the entire major product peak for all targets.”

ii) PgS22 – what is the additional set of ions in the IRAK2 mass spectrum?

The additional masses were identified as sodium (Na⁺) adducts from the LC-MS.

iii) PgsS27, S28, S33 – what are the impurities in the D-NEMO, L-FKBP12, D-YAP1 proteins that are evident from the mass spectra?

The additional masses were identified as sodium (Na⁺) adducts from the LC-MS.

The changes are clearly highlighted in yellow in the enclosed copy of our manuscript. We also provide clean versions of the revised manuscript and supplementary information, and the requested editorial policy checklist, life sciences reporting summary and source data files. With these changes in place, we believe our manuscript is now suitable for publication and we thank you for your consideration of our manuscript.

Sincerely,

Bradley L. Pentelute, Ph.D.

Professor, Department of Chemistry, MIT
Associate Member, Broad Institute of MIT and Harvard
Member, Center for Environmental Health Sciences, MIT
Extramural Faculty, The Koch Institute for Integrative Cancer Research at MIT

REVIEWERS' COMMENTS

Reviewer #1 (Remarks to the Author):

All my previous comments have been addressed. I now recommend publishing this manuscript.

Reviewer #2 (Remarks to the Author):

The authors have sufficiently addressed most of my comments in their revised manuscript and response to reviewers. However, there are a couple of remaining issues that need to be fixed before publication.

- i) The authors have corrected Fig. 4B to now specify that the D-H103 peptide contains norleucine (Nle) rather than methionine. However, the sequence logo for MDM2.C3 in Fig. 4A still includes 'M' rather than Nle. For accessibility to a broader readership, the authors should also define the meaning of Nle as norleucine in the figure legend and/or text.
- ii) I appreciate the authors now including information about the protein:peptide ratio used for crystallisation in section 11 of the supplementary information. However, the final protein concentration used in a crystallisation experiment is a critical factor and this information is still lacking. Therefore, the authors should include this information for each crystallised complex e.g. in the "Crystallisation conditions" row in supplementary tables 11.1.1 – 11.2.4.

Reviewer #3 (Remarks to the Author):

The authors have improved the manuscript significantly following the first review (I was reviewer 2). However, there were points made by both reviewers that I believe warrant the addition of information/clarification in the manuscript itself (not just a rebuttal in a letter). I have detailed the major ones below:

1) The IC50's/Kds for the two previous display studies against MDM2 must be added in the text of the manuscript with some context on why these provided more potent ligands

2) In my previous review I wrote:

"It would be useful for the authors to articulate why these particular D-proteins of the 12 were chosen. The authors describe MDM2 as a benchmarking example but what about CHIP?"

The authors need to state in the manuscript (along the lines of the rebuttal letter) why these were chosen to help the reader understand.

3) point 3 from Reviewer 1 states:

"Many protein target size will go beyond 200 amino acids or 300 amino acids, I would think flow synthesis will still need to be combined with ligation methods to access these bigger target proteins, especially if people want to access these targets with high purity and homogeneity, high quality target preparation is key to the success of final D-peptide discovery, the authors probably should discuss this aspect."

It would be appropriate based on this comment to discuss the limitations of the flow D-protein synthesis in the conclusions of the manuscript, i.e. that ligation chemistry in concert with the technology described in the paper would be necessary for accessing larger mirror image protein targets.

Finally, the authors should comment on why some of the HPLC retention times in Figure 2 for the L- and D-proteins are not identical (barnase, BCL-11, max-max and myc-max look particularly bad). Is this drift on the HPLC instrument? (I assume they were run on the same HPLC system after each other). Again the authors will need to add a footnote in the Figure legend to explain this for the reader

or even better run co-HPLC to show that they do elute together and a true mirror image proteins.
After these points are clarified I would recommend publication.

Bradley L. Pentelute
Professor
Phone: (617) 324-0180
Email: blp@mit.edu

Massachusetts Institute of Technology
Department of Chemistry, Room 18-596
77 Massachusetts Avenue
Cambridge, MA 02139-4307

December 19, 2023

Dear Reviewers,

We thank you for your review of our manuscript entitled “*Mirror-Image Ligand Discovery Enabled by Single-Shot Fast-Flow Synthesis of D-Proteins*” which we submitted to *Nature Communications* (manuscript NCOMMS-23-29181A). We appreciate your efforts and are grateful for your comments. In our point-by-point responses below, the reviewers’ remarks are reproduced in black color, whereas our responses are given in green color.

Reviewer 1 (Remarks to the Author)

All my previous comments have been addressed. I now recommend publishing this manuscript.

We thank the reviewer for their appreciation of our work and their recommendation to publish.

Reviewer #2 (Remarks to the Author)

The authors have sufficiently addressed most of my comments in their revised manuscript and response to reviewers. However, there are a couple of remaining issues that need to be fixed before publication.

i) The authors have corrected Fig. 4B to now specify that the D-H103 peptide contains norleucine (Nle) rather than methionine. However, the sequence logo for MDM2.C3 in Fig. 4A still includes ‘M’ rather than Nle. For accessibility to a broader readership, the authors should also define the meaning of Nle as norleucine in the figure legend and/or text.

We thank the reviewer for pointing out this potential point of confusion. To clarify, the phage display selections against the D-proteins were performed with macrocyclic L-peptide libraries that contained methionine (M), therefore the logo plots are correct in including this amino acid. Methionine was substituted with Nle only at the individual binder synthesis stage, in order to avoid potential oxidation of the Met side-chain. We also revised the figure caption to clarify this point and define Nle and the notations used in describing the sequences as follows: “The individual sequences chosen for validation were synthesized with D-norleucine instead of D-methionine to avoid possible side-chain oxidation reactions. Lower-case letters in the sequences denote single letter abbreviations for D-amino acids, nle = D-norleucine.”

ii) I appreciate the authors now including information about the protein:peptide ratio used for crystallisation in section 11 of the supplementary information. However, the final protein concentration used in a crystallisation experiment is a critical factor and this information is still lacking. Therefore, the authors should include this information for each crystallised complex e.g. in the “Crystallisation conditions” row in supplementary tables 11.1.1 – 11.2.4.

The requested information regarding protein concentrations was added under the appropriate table rows in the revised Supplementary Information Section 3 as requested by the reviewer.

Bradley L. Pentelute
Professor
Phone: (617) 324-0180
Email: blp@mit.edu

Massachusetts Institute of Technology
Department of Chemistry, Room 18-596
77 Massachusetts Avenue
Cambridge, MA 02139-4307

Reviewer #3 (Remarks to the Author)

The authors have improved the manuscript significantly following the first review (I was reviewer 2). However, there were points made by both reviewers that I believe warrant the addition of information/clarification in the manuscript itself (not just a rebuttal in a letter). I have detailed the major ones below:

1) The IC₅₀'s/K_ds for the two previous display studies against MDM2 must be added in the text of the manuscript with some context on why these provided more potent ligands

We provided in the revised manuscript the reported affinity values for the binders discovered in the mentioned studies. Regarding the differences in the reported K_d values, we are hesitant to offer a conclusive explanation. Different libraries (peptide scaffolds) and screening platforms were used in these studies, and some macrocyclic binders were cyclized post-validation, so we cannot directly compare these outcomes with our work other than noting macrocyclic libraries may lead to higher affinity, but an exact affinity range is difficult to predict by just choosing a specific library. In general, macrocyclic scaffolds exhibit structural rigidity, enhanced selectivity and resistance to proteolysis, which contribute to improved stability and binding affinity, as well as longer half-lives.

2) In my previous review I wrote:

"It would be useful for the authors to articulate why these particular D-proteins of the 12 were chosen. The authors describe MDM2 as a benchmarking example but what about CHIP?"

The authors need to state in the manuscript (along the lines of the rebuttal letter) why these were chosen to help the reader understand.

MDM2 was indeed selected as a well characterized protein by phage display and availability of data on MIPD protocols. CHIP was selected because it is also an E3 ubiquitin-protein ligase, therefore in the same family as MDM2, and for its potential therapeutic relevance by being implicated in neurological disease. We revised the respective section of the main text to explain our choice.

3) point 3 from Reviewer 1 states:

"Many protein target size will go beyond 200 amino acids or 300 amino acids, I would think flow synthesis will still need to be combined with ligation methods to access these bigger target proteins, especially if people want to access these targets with high purity and homogeneity, high quality target preparation is key to the success of final D-peptide discovery, the authors probably should discuss this aspect."

It would be appropriate based on this comment to discuss the limitations of the flow D-protein synthesis in the conclusions of the manuscript, i.e. that ligation chemistry in concert with the technology described in the paper would be necessary for accessing larger mirror image protein targets.

Bradley L. Pentelute

Professor

Phone: (617) 324-0180

Email: blp@mit.edu

Massachusetts Institute of Technology

Department of Chemistry, Room 18-596

77 Massachusetts Avenue

Cambridge, MA 02139-4307

We have added a section regarding limitations of the flow synthesis technology to the discussion section as follows: “We acknowledge that individual optimization may be needed for the flow synthesis of specific protein sequences, identified in our laboratory on a case-by-case basis. This was not the case here, however, as each target protein was synthesized successfully on the initial attempts according to our established protocol.

In our work, these mirror-image proteins were rapidly prepared in a single shot for screening in a standardized format and the ease of synthesis benefitted from their relatively small size (<150 amino acids). Making MIPD routinely applicable to larger protein targets may likely require combining flow synthesis with ligation methods. Our ongoing efforts are focused on leveraging these two powerful synthetic approaches.”

Finally, the authors should comment on why some of the HPLC retention times in Figure 2 for the L- and D-proteins are not identical (barnase, BCL-11, max-max and myc-max look particularly bad). Is this drift on the HPLC instrument? (I assume they were run on the same HPLC system after each other). Again the authors will need to add a footnote in the Figure legend to explain this for the reader or even better run co-HPLC to show that they do elute together and a true mirror image proteins.

The HPLC chromatograms for the proteins in question were indeed acquired on different days, accounting for the variations in retention time, and the purified D-variants were taken directly into biological activity studies. A footnote was added to the figure caption to describe this outcome.

After these points are clarified I would recommend publication.

We thank the reviewer for their comments and support for our work.

The changes are clearly highlighted in yellow in the enclosed copy of our manuscript. We also provide clean versions of the revised manuscript and supplementary information, and the requested editorial policy checklist, life sciences reporting summary and source data files. With these changes in place, we believe our manuscript is now suitable for publication and we thank you for your consideration of our manuscript.

Sincerely,

Bradley L. Pentelute, Ph.D.

Professor, Department of Chemistry, MIT

Associate Member, Broad Institute of MIT and Harvard

Member, Center for Environmental Health Sciences, MIT

Extramural Faculty, The Koch Institute for Integrative Cancer Research at MIT